# Pregnancy and a high-fat, high-sugar diet each attenuate mechanosensitivity of murine gastric vagal afferents, with no additive effects

Georgia S. Clarke[1,2,3] , Hui Li[1,3], Elaheh Heshmati[1,3], Lisa M. Nicholas[1,2,4], Kathryn L. Gatford[1,2,3] and Amanda J. Page[1,3]

[1]*School of Biomedicine, The University of Adelaide, Adelaide, South Australia, Australia*

[2]*Robinson Research Institute, The University of Adelaide, Adelaide, South Australia, Australia*

[3]*Nutrition, Diabetes & Gut Health, Lifelong Health Theme, South Australian Health and Medical Research Institute, SAHMRI, Adelaide, South Australia, Australia*

[4]*Adelaide Centre for Epigenetics, The University of Adelaide, Adelaide, South Australia, Australia*

Handling Editors: Kim Barrett & Michel Neunlist

The peer review history is available in the Supporting Information section of this article (https://doi.org/10.1113/JP286115#support-information-section).

**Abstract figure legend** Pregnancy and a high-fat, high-sugar diet each attenuate mechanosensitivity of murine gastric vagal afferents (GVAs), with no additive effects. Illustration of the effect of pregnancy and a high-fat, high-sugar (HFHS) diet on GVA satiety signalling and the impact on food intake. Tension-sensitive GVA responses to stretch are attenuated in late pregnant mice and non-pregnant mice fed an HFHS diet. This is associated with an increase in food intake. Tension-sensitive GVA responses to stretch are attenuated in late pregnant mice fed an HFHS diet, but with no additive effect. Similar to the late pregnant or non-pregnant HFHS-fed mice, there was an increase in food intake in the late-pregnant HFHS-fed mice.

**Abstract** Gastric vagal afferents (GVAs) sense food-related mechanical stimuli and signal to the CNS to initiate meal termination. Pregnancy and diet-induced obesity are independently associated with dampened GVA mechanosensitivity and increased food intake. Whether a high-fat, high-sugar diet (HFHSD) impacts pregnancy-related adaptations in GVA signalling is unknown and was investigated in this study. Three-week-old female Glu Venus-expressing mice, on a C57BL/6 background, were fed standard laboratory diet (SLD) or HFHSD for 12 weeks, and then half of each group were mated to generate late pregnant (Day 17.5; P-SLD $N = 12$, P-HFHSD $N = 14$) or non-pregnant (NP-SLD $N = 12$, NP-HFHSD $N = 16$) groups. Body weight and food intake were monitored in Promethion metabolic cages from before mating until Day 17.5 of pregnancy or equivalent ages in non-pregnant mice, prior to tissue collection at 07.00 h for *in vitro* single fibre GVA recording and gene expression analysis. Pregnant mice gained more weight than non-pregnant mice but weight gain was unaffected by diet. By mid-pregnancy, light-phase food intake (kJ and g) was higher in pregnant than in non-pregnant mice (each $P < 0.001$) due to larger meals (kJ and g, each $P < 0.001$), irrespective of diet. Pregnancy and HFHSD-feeding reduced tension-sensitive GVA mechanosensitivity (each $P < 0.01$), but pregnancy did not further downregulate GVA stretch responses within HFHSD mice ($P = 0.652$). Nodose ganglia growth hormone receptor mRNA abundance was upregulated in pregnancy, possibly contributing to lower GVA mechanosensitivity during pregnancy in SLD mice. Larger light-phase meals in pregnant compared to non-pregnant HFHSD mice may therefore reflect the downregulation of other satiety pathways.

(Received 10 December 2023; accepted after revision 3 February 2025; first published online 3 March 2025)

**Corresponding author** A. J. Page: Vagal Afferent Research Group, School of Biomedicine, University of Adelaide, Level 7 South Australian Health and Medical Research Institute, Adelaide, South Australia 5000, Australia. Email: Amanda.page@adelaide.edu.au

## Key points

- Gastric vagal afferents (GVAs) regulate food intake by sensing the arrival and quantity of food and communicating this information to the brain.
- In standard laboratory diet (SLD) mice, gastric tension-sensitive vagal afferent mechanosensitivity was attenuated in pregnant compared to non-pregnant mice, which is concurrent with increases in total food intake and meal size.
- Nodose ganglia growth hormone receptor mRNA abundance was increased in pregnancy, possibly accounting for attenuated GVA mechanosensitivity in pregnant SLD mice.
- In non-pregnant mice, tension-sensitive GVA mechanosensitivity was selectively attenuated in high-fat, high-sugar diet (HFHSD) compared to SLD mice. Despite this, HFHSD mice ate less food and smaller meals compared to the SLD mice, suggesting other satiety mechanisms are limiting food intake.
- Despite higher food intake, there was no further reduction in mechanosensitivity in pregnant HFHSD mice compared to non-pregnant HFHSD mice and further studies are required to increase understanding of food intake regulation across pregnancy.

**Georgia S. Clarke** is a recent PhD graduate in the School of Biomedicine at the University of Adelaide. She is interested in pregnancy, nutrition and neuroscience, which were all encapsulated within her research project, which focused on understanding adaptations in gastrointestinal satiety signalling during pregnancy and the role of pregnancy hormones in driving these changes.

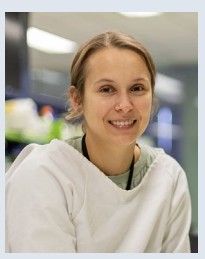

# Introduction

Food intake is highly regulated and remains relatively stable during steady-state conditions (Cummings & Overduin, 2007), whilst highly plastic regulatory mechanisms allow food intake to adapt rapidly to changing metabolic demands. The gastrointestinal tract (GIT) plays a key role in the regulation of food intake by sensing food intake and signalling via vagal afferents to the CNS to modulate satiety (Cummings & Overduin, 2007). Gastrointestinal vagal afferents (GVAs) are sensory fibres located primarily in the stomach and intestinal wall that detect the arrival, volume and chemical composition of a meal (Page et al., 2002). In the stomach, GVAs primarily respond to mechanical stimuli. Mechanosensitive tension-sensitive GVAs respond to distension following meal intake and are thought to act centrally to induce satiation (Page et al., 2002) and also as feedback signals to regulate gut function such as gastric accommodation and motility (Li & Page, 2022). The other subtype of mechanosensitive GVAs is mucosal afferents, which respond to mucosal stroking and are thought to detect particle density and regulate gastric emptying (Page et al., 2002). GVAs are highly plastic, responding to circadian cues and nutritional status (Kentish, Frisby, et al., 2013). For example, tension-sensitive GVA responses to stretch are attenuated after fasting (Kentish et al., 2012), which is consistent with the increase in the size of the first meal after a fast (Kentish et al., 2012; Le Magnen et al., 1980). Furthermore, GVA signalling adapts to allow changes in food intake in response to long-term changes in energy demand, such as in pregnancy, where increased maternal energy intake is required to support maternal adaptations and increased metabolic rate, fetal and placental growth and to prepare for future lactation (Li et al., 2021). To meet these demands, daily food intake increases by around 200–300 calories in the third trimester in pregnant women and by around 25% from mid-pregnancy onwards in mice (Clarke et al., 2021). In parallel with increasing food intake, GVA responses to distension are attenuated from mid-pregnancy onwards, allowing greater food intake before induction of satiety signals, compared to non-pregnant mice (Li et al., 2021). Indeed, the same pregnant mice display altered eating behaviours, with the consumption of larger meals over a longer meal duration during the light-phase (Li et al., 2021). Interestingly, tension-induced signalling by murine gastric (Kentish et al., 2012) and jejunal vagal afferents (Daly et al., 2011) are also attenuated in high-fat diet (HFD)-induced obesity, with reduced signalling likely to promote increased food intake. Similar to pregnancy, HFD mice also exhibit altered feeding patterns, with an increase in energy consumption and meal number during the light-phase compared to those fed standard chow, suggesting their inability to sense satiation signals

(Kentish et al., 2016). In developed countries, increasing rates of obesity are occurring in the context of diets that are high in both fat and sugar. For example, in countries including the USA, Australia, New Zealand and parts of Europe, 70% of calories arise from animal foods, oils, fat and sweeteners (Clemente-Suárez et al., 2023). The impacts of a high-fat, high-sugar diet (HFHSD) on GVA responses to mechanical food-related stimuli have not been reported to date.

The combination of obesity and pregnancy is increasingly common. Almost 50% of women are overweight or obese prior to pregnancy (Australian Institute of Health Welfare, 2022; Wang et al., 2021). Obesity during pregnancy increases the risks of short-term complications and predisposes offspring to metabolic diseases in later life (Poston et al., 2011). In addition, more than 50% of women experience excessive weight gain during pregnancy (Deputy et al., 2015), which is itself associated with increased risks of complications including gestational diabetes, the need for caesarean-section delivery and infant macrosomia (Goldstein et al., 2018). Excessive weight gain during pregnancy may, in part, reflect the impacts of obesity on appetite-regulatory pathways. In both lean and overweight/obese women, pregnancy is associated with altered main meal patterns, with increasing meal frequency and snack-dominant meal patterns as gestation progresses (Ainscough et al., 2020). However, greater weight gain in obese than lean pregnant women may be attributed to diet composition, as they consume a diet higher in processed foods and confectionary snacks (Flynn et al., 2016; Lindsay et al., 2015). Furthermore, the increase in energy intake during pregnancy is higher in Western diet-fed than standard chow-fed mice (King et al., 2013; Samuelsson et al., 2008). Given that obese women are entering pregnancy at an increased risk of pregnancy complications, strategies to reduce energy intake or enable adherence to nutritional guidelines are clinically important. The mechanisms permitting the overconsumption of food during HFHSD feeding in pregnancy are unknown, including possible changes in GVA signalling. We investigated this question using a mouse model of HFHSD feeding, focusing on late pregnancy when the depression of GVA responses to stretch is greatest (Li et al., 2021).

# Materials and methods

### Ethical approval

All studies were approved by the Animal Ethics Committee (SAM-21-048) of the South Australian Health and Medical Research Institute (SAHMRI) and carried out in accordance with the Australian code for the care and use of animals for scientific purposes, 8th edition

2013 and adhere to the Arrive 2.0 guidelines (du Sert et al., 2018; Grundy, 2015).

## Animals and experimental design

Glu Venus-expressing mice (Reimann et al., 2008), maintained on a C57BL/6 background, were obtained under a material transfer agreement from Cambridge Enterprise Limited, Cambridge, UK, and bred at the SAHMRI bioresources facility. The mice in this study were part of a larger study where the use of this strain allowed the separation of $\alpha$- and $\beta$-cells within pancreatic cell populations. Mice were housed at 22°C, in a 12:12 h light/dark cycle with lights on at 07.00 h. Female mice (3–4 weeks old, 9–20 g) were randomised, using a simple table method, to be fed a standard laboratory diet (SLD, $N = 24$: Teklad standard diet: 13 kJ/g, digestible energy from protein 24%, fat 18% and carbohydrates 58%, CAT no.: 2018, Envigo, Cambridge, UK) or an HFHSD [$N = 30$: Specialty Feeds: 23 kJ/g, digestible energy from protein 17.6%, fat 58.4% (derived from soya bean and coconut oil) and carbohydrates 24% (sucrose 175 g/kg), CAT no.: SF21-003, Glen Forrest, Western Australia, Australia) for 12 weeks (diet phase). Mice were housed in groups of 2–5 litter mates and weighed weekly during the feeding phase of the study, from weaning until 14–15 weeks of age. Mice were then single-housed in metabolic cages for a 7 day acclimatisation period (Promethion Sable System, Las Vegas, NV, USA). Small wooden sticks were placed in every cage during the feeding phase and whilst in the metabolic cages to allow for the wearing down of teeth. Following acclimatisation, female mice were randomised to mating, using a simple table method and pair-housed at 17.00 h with a male mouse in a home cage for mating, remaining on their diets during the mating period. Female mice were checked daily at 07.00 h, and pregnancy was confirmed by the presence of a vaginal plug (assigned as Day 0.5 of pregnancy). Plugged females were then returned to individual metabolic cages until the late-pregnancy endpoint at Day 17.5 (SLD $N = 12$, HFHSD $N = 14$). Control (non-pregnant) female mice (SLD $N = 12$, HFHSD $N = 16$) were pair-housed with another female in a normal home cage and returned to metabolic cages on age-matched days. Mice were kept on respective diets during 17.5 day period. On Day 17.5, mice were anaesthetised between 07.00 and 07.30 h by isoflurane inhalation (5% in oxygen) and humanely killed by decapitation prior to tissue collection for electrophysiology experiments described below. Maternal gonadal and perirenal fat pads and individual fetuses were dissected and weighed.

Mice that were mated, with vaginal plugs present, but did not become pregnant ($N = 11$) were excluded from the study and not included in the final mating or pregnancy numbers. These mice were not added to the non-pregnant group, to avoid the potential impacts of elevated prolactin during pseudo-pregnancy (Phillipps et al., 2020). One non-pregnant mouse was excluded from the study due to over-barbering and another was unexpectedly found dead in the cage. The planned sample size of $N = 12$ was based on variation in GVA function in previous studies within our laboratory (Kentish et al., 2012; Li et al., 2021).

## Metabolic monitoring

Metabolic cages were used to continuously measure body weight and record real-time feeding events, including total food intake, average meal size and duration and total meal number, and analysed as previously described (Li et al., 2021). Briefly, metabolic data were transformed using the Promethion data software package ExpeData version 1.9.14 (Promethion Sable System) using analytical macro 6. Data from each day of the study were divided into 12 h time periods corresponding to the light- and dark-phases. Body weight is presented for each study day (averaged across 24 h) and food intake parameters (food and energy intake, meal size in energy and grams, meal duration, meal number) are presented as averages across two gestational days or age-matched days. All mice were included in the analysis, but data points were excluded if they did not include a full 12 h of data for each photo-period, for example due to cage changes.

## *In vitro* mouse GVA electrophysiology

The electrophysiological methods used to record mouse GVA activity have been described in detail previously (Li et al., 2013; Page et al., 2002). Briefly, the thorax was opened to remove the stomach and oesophagus and the vagal nerves were separated from the oesophagus. The stomach was opened with the vagal nerves attached and placed mucosal side up in an organ bath filled with a modified Krebs solution, including nifedipine (1 μM) to prevent smooth muscle contraction. The vagal nerves were placed into another chamber filled with liquid paraffin. The nerves were teased apart into small bundles and placed onto a platinum recording electrode for single-fibre recording. Nerve impulses were amplified (DAM50, World Precision Instruments, Sarasota, FL, USA), filtered (Band-pass filter 932, CWE, Ardmore, PA, USA) and recorded.

GVA mechanosensitivity was identified by locating receptive fields on the stomach, where tension-sensitive GVAs respond to mucosal stroking and tension stimuli, whilst mucosal GVAs respond to mucosal stroking only (Page et al., 2002). To record the responses of tension-sensitive afferents to stretch, a threaded hook was attached adjacent to the receptive field and to a cantilever

system. Tension stimuli were created by placing weights (0.5–5 g) on the cantilever system for 1 min. To record the responses of mucosal afferents, the receptive field was stroked with calibrated von Frey hairs (10–1000 mg). Up to five individual tension-sensitive or mucosal afferents were recorded per mouse. Action potentials of single units were analysed using Spike 2 software (Cambridge Electronic Design, Cambridge, UK). When recordings were obtained from more than one GVA subtype in an individual mouse, data were averaged to create one data point per GVA subtype per mouse.

### Nodose ganglia quantitative RT-PCR

Whole nodose ganglia were removed bilaterally and snap-frozen. Total RNA was extracted using an Invitrogen PureLink RNA Micor Kit (Life Technologies, Mulgrave, Australia), according to the manufacturer's instructions. Total RNA quality was assessed using the NanoDrop Lite spectrophotometer (Thermo Fisher Scientific, Adelaide, Australia), estimated by the $A_{260/280}$ ratio. Qualitative real-time PCR (qRT-PCR) was conducted on a 7500 Fast thermocycler (Applied Biosystems® 7500 Real-Time PCR System, Thermo Fisher Scientific) using TaqMNan™ EXPRESS One-Step Superscript RT-PCR kits (Invitrogen, Carlsbad, CA, USA). Assays were predesigned for transient receptor potential vanilloid 1 (*TRPV1*; Assay ID: Mm01246302_m1), cholecystokinin A receptor (*CCKA*; Assay ID: Mm00438060_m1), growth hormone receptor (*GHR*; Assay ID: Mm00439093_m1), leptin receptor (*LepR*; Assay ID: Mm00440181_m1), growth hormone secretagogue receptor [*GSHR* (ghrelin receptor); Assay ID: Mm00616415_m1], $\beta$-actin [*ACTB* (housekeeper); Assay ID: MM02619580_g1] and beta-2-microglobulin [*B2M* (housekeeper), Assay ID: Mm00437762_m1]. The housekeeping genes, *ACTB* and *B2M*, were chosen based on their stability across samples determined from normfinder software (Christie et al., 2020), with a stability value of 0.004. Each assay was run in duplicate, and transcript levels were calculated relative to the averaged cycle threshold (Ct) value of reference genes using the $2^{-\Delta CT}$ method (Bookout & Mangelsdorf, 2003).

### Statistical analysis

All data are presented as mean $\pm$ SD with $N$ = number of animals. Statistical analyses were conducted using SPSS v. 28 (IBM Corp., Armonk, NY, USA). Effects of diet on body weight and weight gain (Weeks 1–12) were analysed by one-way ANOVA. Body weight during acclimatisation (Week 13) and the 17 day period from mating (and the equivalent period in non-pregnant age-matched controls) was analysed by the linear mixed model to assess the effect of pregnancy (pregnant *vs.* non-pregnant) and diet (SLD

*vs.* HFHSD) with the day as a repeated factor. Where a pregnancy $\times$ diet $\times$ day interaction was significant, mixed repeated models were used to assess the effects of diet and day separately in non-pregnant and pregnant groups, and the effects of pregnancy and day separately within SLD and HFHSD mice. Where diet $\times$ day and/or pregnancy $\times$ day interactions were significant, a two-way ANOVA was used to assess the effects of pregnancy and diet separately for each day of the study. Effects of pregnancy and diet on body weight gain (acclimatisation – Day 17), and fat pad weights (gonadal and perirenal) were analysed by two-way ANOVA. Where diet $\times$ pregnancy interactions were significant, the effects of pregnancy within each diet group and the effects of diet within non-pregnant and pregnant groups were analysed by one-way ANOVA. Effects of diet on litter size and average pup weight were analysed by one-way ANOVA.

Full-day, light- and dark-phase food intake parameters were analysed using linear mixed models to assess the effect of pregnancy (pregnant *vs.* non-pregnant) and diet (SLD *vs.* HFHSD) with day as a repeated factor, as described above for body weight analyses. Because the diets differed in composition, we analysed food intake and meal size in terms of both weight and energy content. We also separately analysed the effects of pregnancy and diet on average food intake and meal size over the final two study days (closest to the time of GVA assessment), using a two-way ANOVA. Full-day, light- and dark-phase food intake parameters were included and where a diet $\times$ pregnancy interaction was significant, we then analysed the effects of pregnancy within each diet group and the effects of diet within each pregnancy group using one-way ANOVA.

The mechanosensitivity of gastric tension-sensitive and mucosal afferents was analysed using a linear mixed model to assess the effect of pregnancy (pregnant *vs.* non-pregnant) and diet (SLD *vs.* HFHSD), and with load [circular tension (g) or von Frey hair (mg), respectively] as a repeated factor. Where a pregnancy $\times$ diet $\times$ load interaction was significant, we used mixed repeated models to assess effects of diet and load separately within each pregnancy group, and to assess effects of pregnancy and load separately within each diet group. Where diet $\times$ load and/or pregnancy $\times$ load interactions were significant, we ran a two-way ANOVA for each level of load, to assess the effects of pregnancy and diet. Where a diet $\times$ pregnancy interaction was significant for a given load, we used one-way ANOVA to assess the effects of pregnancy within each diet group and of diet within each pregnancy group. To determine if there was a correlation between GVA sensitivity and meal size, the responses to GVA tension (5 g) and mucosal (200 mg) afferents, recorded during the light-phase, were plotted against meal size during the light-phase and a Pearson correlation was performed.

**Table 1. Mouse phenotype**

| Diet phase | Treatment groups | | | | | | *P* (ANOVA) |
|---|---|---|---|---|---|---|---|
| | SLD (*N* = 24) | | | HFHSD (*N* = 30) | | | Diet |
| Body weight at week 0 (g) | 15.4 ± 2.1 | | | 14.8 ± 2.1 | | | 0.245 |
| Body weight at week 12 (g) | 22.5 ± 2.1 | | | 23.9 ± 2.8 | | | 0.059 |
| Weight gain, in diet phase (g) | 7.1 ± 1.9 | | | 9.1 ± 2.4 | | | 0.002 |
| End of study | NP-SLD (*N* = 12) | P-SLD (*N* = 12) | NP-HFHSD (*N* = 16) | P-HFHSD (*N* = 14) | Diet | Pregnancy | Diet × pregnancy |
| Body weight (g) | 24.1 ± 2.4 | 34.8 ± 2.6 | 27.3 ± 4.1 | 35.2 ± 2.8 | 0.046 | <0.001 | 0.127 |
| Gonadal fat mass (g) | 0.35 ± 0.22 | 0.34 ± 0.08 | 0.65 ± 0.48 | 0.48 ± 0.26 | 0.014 | 0.271 | 0.345 |
| Perirenal fat mass (g) | 0.11 ± 0.09 | 0.26 ± 0.13[a] | 0.29 ± 0.25[b] | 0.21 ± 0.11 | 0.219 | 0.445 | 0.025 |
| Litter size (*N*) | N/A | 7.3 ± 2.3 | N/A | 6.6 ± 2.2 | 0.395 | – | – |
| Average pup weight (g) | N/A | 0.87 ± 0.12 | N/A | 0.86 ± 0.15 | 0.836 | – | – |

Mice were fed either a standard laboratory diet (SLD) or high-fat high-sugar diet (HFHSD) for 12 weeks from weaning, then randomised to mating (pregnant group, P) or to be unmated controls (non-pregnant group, NP), and remained on their diets for a further 17 days from mating or age-matched days. Data are mean ± SD. N/A, not applicable; NS, not significant. The effects of diet on body weights at Weeks 0 and 12 and weight gain were analysed using a one-way ANOVA. The effects of pregnancy and diet on body weight at Day 17 and fat pad weights (gonadal and perirenal) were analysed using a two-way ANOVA. Where diet × pregnancy interactions were significant, one-way ANOVAs were used to assess the effects of pregnancy within each diet group, and the effects of diet within non-pregnant and pregnant groups. The ffects of diet on litter size and average pup weight was analysed by one-way ANOVA. [a]Within SLD mice, perirenal fat mass was heavier in P than NP mice (*P* = 0.003); [b]within NP-mice, NP-HFHSD mice had heavier perirenal fat mass than NP-SLD mice (*P* = 0.035).

The effects of pregnancy and diet on transcript expression were analysed by two-way ANOVA.

## Results

### Phenotype

The body weight of mice was not different between SLD and HFHSD groups prior to starting the diet (Table 1). After 12 weeks on the diet, body weights of HFHSD and SLD mice were similar, but HFHSD mice gained more weight than SLD mice (Table 1). From acclimatisation onwards, the effects of pregnancy and diet on body weight differed with day of study (three-way interactions, *P* < 0.001, Fig. 1*A*). HFHSD mice were heavier than SLD mice from acclimatisation (Week 13) until the end of the study, except on Day 15 (all *P* < 0.05, Fig. 1*A*). Pregnant mice were heavier than non-pregnant mice from Day 9 onwards (Fig. 1*A*). Effects of diet on total weight gain differed between pregnant and non-pregnant mice (diet × pregnancy interaction, *P* = 0.027, Fig. 1*B*). Not surprisingly, pregnant mice gained more weight than non-pregnant mice within both diet groups (Fig. 1*B*). Diet did not affect weight gain within pregnant mice, whilst NP-HFHSD mice gained more weight than NP-SLD mice (*P* = 0.016, Fig. 1*B*).

At the end of the study, the gonadal fat pad was heavier in HFHSD than SLD mice independent of pregnancy (Table 1). The effect of pregnancy on perirenal fat differed between diets (diet × pregnancy interaction, *P* = 0.025, Table 1). The perirenal fat pad was heavier in NP-HFHSD compared to NP-SLD mice but was not affected by maternal diet within pregnant mice. Perirenal fat pad weight was also increased in P-SLD compared to NP-SLD mice. This effect was not present in HFHSD mice (Table 1). Maternal diet had no effect on litter size and average pup weights (Table 1).

## Impacts of diet and pregnancy on food intake behaviours

Table 2 provides a summary of all 24 h, light- and dark-phase food intake data.

**Energy intake.** Across a 24 h period, the effects of diet on energy intake (kJ) differed with both day (day × diet interaction: $P = 0.038$) and pregnancy (pregnancy × diet interaction: $P = 0.017$). We therefore analysed the effects of diet and pregnancy for each 2 day period throughout the study (Fig. 2*Ai*). HFHSD mice consumed more energy than SLD mice between Days 0.5–8.5 and 12.5–14.5 (all $P < 0.05$, Fig. 2*Ai*), but there was no effect of pregnancy (all $P > 0.05$). On Days 8.5–10.5 and 12.5–14.5, effects of diet depended on pregnancy status (pregnancy × diet interactions: each $P < 0.05$). On both days, NP-HFHSD mice consumed more energy than NP-SLD mice (both $P < 0.01$, Fig. 2*Ai*). Pregnancy did not affect food intake on Days 8.5–10.5 and 12.5–14.5 within each diet group and diet did not affect food intake within pregnant mice (all $P > 0.05$).

During the light-phase, effects of diet on energy intake differed with both day (day × diet interaction: $P = 0.035$) and pregnancy (pregnancy × diet interaction: $P < 0.001$). Therefore, we analysed the effects of diet and pregnancy for each 2 day period throughout the study (Fig. 2*Aii*). Energy intake was greater in SLD than HFHSD mice during acclimatisation only ($P < 0.05$, Fig. 2*Aii*). Irrespective of diet, light-phase energy intake was greater

in pregnant than non-pregnant mice, throughout Days 8.5–12.5 and 14.5–17.5 (all $P < 0.05$, Fig. 2*Aii*).

During the dark-phase, the effects of pregnancy and diet on energy intake differed between days (day × pregnancy × diet interaction: $P = 0.043$, Fig. 2*Aiii*), and we therefore analysed the effects of diet and pregnancy for each 2 day period throughout the study. From acclimatisation until Day 8.5, energy intake was greater in the HFHSD than SLD mice (all $P < 0.05$), and did not differ between pregnant and non-pregnant mice (Fig. 2*Aiii*). Neither diet nor pregnancy status affected energy intake on Days 10.5–12.5 or 16.5–17.5 (Fig. 2*Aiii*). The effects of pregnancy on energy intake during the dark-phase differed between diets on Days 8.5–10.5, 12.5–14.5 and 14.5–16.5 (pregnancy × diet interaction: each $P < 0.05$, Fig. 2*Aiii*). On Days 8.5–10.5 and 12.5–14.5, energy intake was greater in NP-HFHSD than P-HFHSD mice (all $P < 0.05$) and was not affected by pregnancy status in SLD mice (Fig. 2*Aiii*). Conversely, on Days 14.5–16.5, energy intake was greater in pregnant than non-pregnant SLD mice ($P < 0.05$) and not affected by pregnancy status in HFHSD mice (Fig. 2*Aiii*). On Days 8.5–10.5, 12.5–14.5 and 14.5–16.5, energy intake was greater in NP-HFHSD compared to NP-SLD mice (all $P < 0.05$) and was not affected by diet in pregnant mice (Fig. 2*Aiii*).

**Food intake.** Within a 24 h period, the effects of diet on food intake (g) differed with both day (day × diet

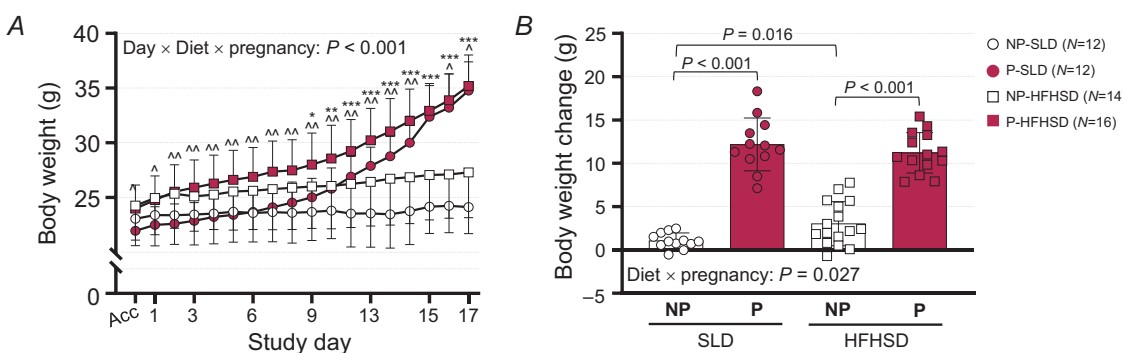

**Figure 1. Impact of diet and pregnancy on body weight**
*A*, daily body weight from acclimatisation to Day 17.5 of non-pregnant (NP) and pregnant (P) mice fed a standard laboratory diet (SLD; non-pregnant, NP, open circles: *N* = 12; pregnant, P, closed circles: *N* = 12) or high-fat high-sugar diet (HFHSD; NP, open squares: *N* = 16; P, closed squares: *N* = 14) for 12 weeks prior and during the study period. Data are presented as mean ± SD and were analysed using a linear mixed model to assess the effect of pregnancy (pregnant *vs*. non-pregnant) and diet (SLD *vs*. HFHSD) with day as a repeated factor. Because the pregnancy × diet × day interaction was significant, two-way ANOVAs were run to assess the effects of pregnancy and diet on each day of the study. Diet effect represented by ^*P* < 0.05, ˜*P* < 0.01, ˜˜*P* < 0.001 and pregnancy effect by **P* < 0.05, ***P* < 0.01, ****P* < 0.001. Exact *P*-values are reported in Table S1 on figshare (DOI: https://doi.org/10.25909/28079570). *B*, change in body weight of P- and NP-SLD mice and P- and NP-HFHSD mice from acclimatisation to Day 17.5 of the study. Bars and whiskers show mean ± SD, with data from each mouse indicated by symbols. Data were analysed using a two-way ANOVA. Where diet × pregnancy interactions were significant, one-way ANOVAs were used to assess the effects of pregnancy within each diet group, and the effects of diet within non-pregnant and pregnant groups.

**Table 2. Summary of the impact of diet and pregnancy on food intake behaviours across 24 h and the light- and dark-phases**

| Time of day | | | | | | |
|---|---|---|---|---|---|---|
| | 24 h | | Light-phase | | Dark-phase | |
| Energy intake (kJ) | 0.5–8.5 | HFHSD > SLD | 8.5–12.5, 14.5–17.5 | P > NP | Acclim–8.5 | HFHSD > SLD |
| | 8.5–10.5 and 12.5–14.5 | In NP: SLD > HFHSD | | | 8.5–10.5, 12.5–14.5: | In HFHSD: NP > P In NP: HFHSD > SLD |
| | | | | | 14.5–16.5 | In SLD: P > NP In NP: HFHSD > SLD |
| Food intake (g) | Acclim, 2.5–6.5, 10.5–12.5, 16.5–17.5 | SLD > HFHSD | All days | SLD > HFHSD | All days | In P: SLD > HFHSD In SLD: P > NP In HFHSD: P < NP |
| | 6.5–10.5, 12.5–16.5 | In P: SLD > HFHSD | 8.5–10.5 onwards | SLD > HFHSD | | |
| | 12.5–16.5 | In NP: SLD > HFHSD | | | | |
| | 14.5–16.5 | In SLD: P > NP | | | | |
| Energy per meal (kJ) | All days | HFHSD > SLD | 2.5–6.5, 8.5–12.5, 14.5–17.5 | P > NP | Acclim–2.5, 6.5–8.5 | HFHSD > SLD |
| | | | 0.5–2.5, 8.5–10.5, 14.5–16.5 | HFHSD > SLD | 14.5–16.5 | In SLD: P > NP |
| Meal size (g) | Acclim–6.5, 8.5–12.5 and 16.5–17.5 | SLD > HFHSD | All days except 2.5–4.5, 14.5–16.5 | SLD > HFHSD | All days except 14.5–16.5 | SLD > HFHSD |
| | 6.5–8.5 and 12.5–16.5 | In P: SLD > HFHSD | 2.5–4.5 | In NP: SLD > HFHSD In P: SLD > HFHSD In SLD: P > NP | 14.5–16.5 | In NP: SLD > HFHSD In P: SLD > HFHSD In SLD: P > NP |
| | 14.5–16.5 | In NP: SLD > HFHSD In SLD: P > NP | 14.5–16.5 | In NP: SLD > HFHSD In P: SLD > HFHSD In SLD: P > NP In HFHSD: P > NP | | |
| Meal duration (min) | All days | SLD > HFHSD P > NP Differed with day | All days | SLD > HFHSD | All days: | In NP: SLD > HFHSD In P: SLD > HFHSD In SLD: NP > P In HFHSD: NP > P |
| Meal number (N) | All days | Differed with day | Acclim | SLD > HFHSD | 2.5–4.5–6.5, 12.5–14.5, 16.5–17.5 | HFHSD > SLD |
| | | | 14.5–16.5 | In P: HFHSD > SLD | 0.5–2.5 | In NP: HFHSD > SLD. |
| | | | | | 8.5–10.5: | In NP: HFHSD > SLD In HFHSD: NP > P |

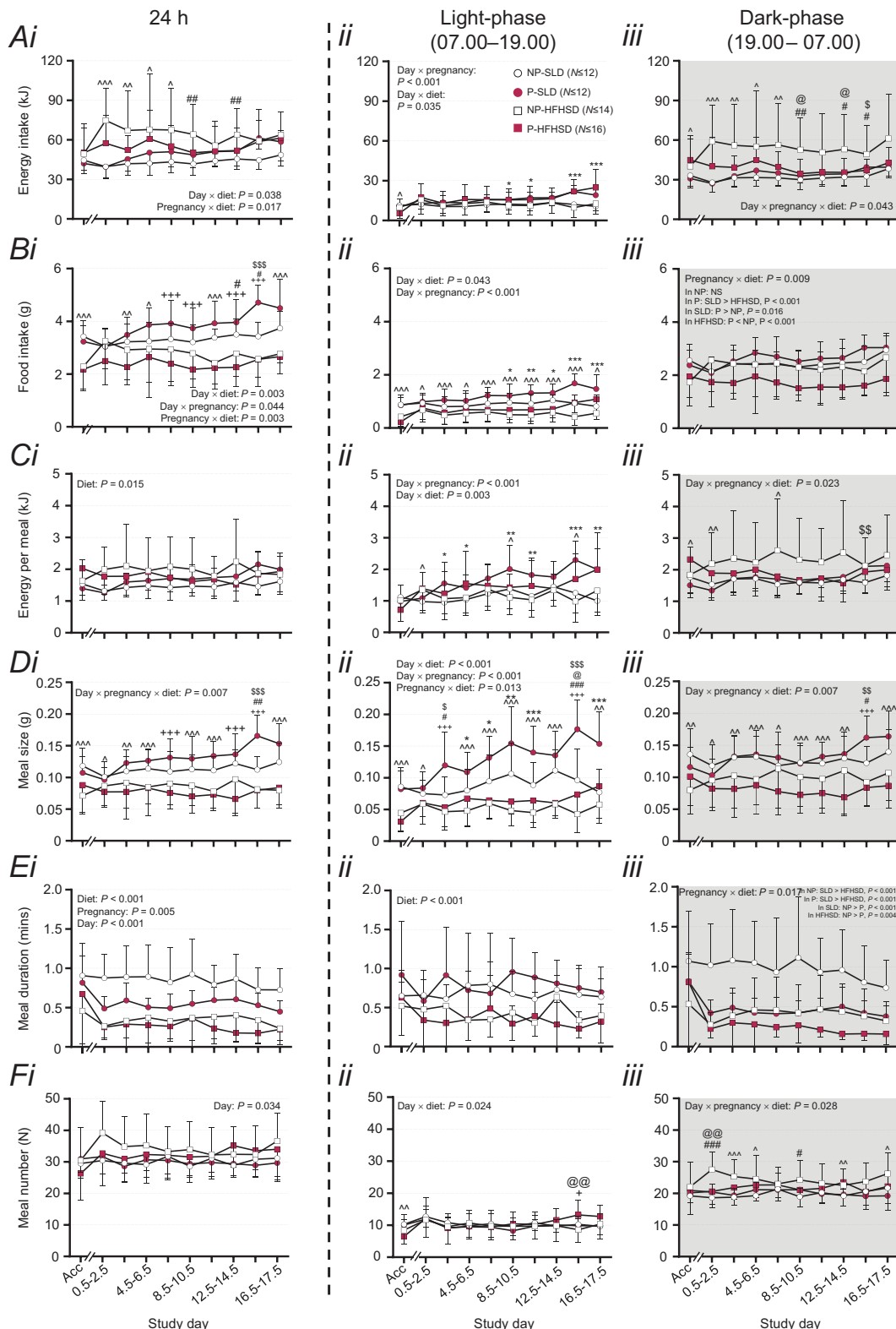

**Figure 2. Impacts of diet and pregnancy on food intake behaviours**
Food intake behaviours of pregnant (P) and non-pregnant (NP) mice exposed to a standard laboratory diet (SLD, non-pregnant, NP, open circles: $N \leq 12$; pregnant; P, closed circles: $N \leq 12$) and high-fat high-sugar diet (HFHSD, NP, open squares: $N \leq 16$; P, closed squares: $N \leq 14$). Total food intake in energy content (kJ, *Ai, ii* and *iii*) and grams (g, *Bi, ii* and *iii*), meal size in energy content (kJ, *Ci, ii* and *iii*) and grams (g, *Di, ii* and *iii*), meal number (*Ei, ii, iii*) and meal

duration (*Fi*, *ii* and *iii*) across 24 h, (*i*) light-phase (*ii*) and dark-phase (*iii*, shaded). Data are presented as mean ± SD, and were analysed using linear mixed models to assess the effect of pregnancy (pregnant *vs.* non-pregnant) and diet (SLD *vs.* HFHSD) with day as a repeated factor. Where a pregnancy × diet × day interaction was significant, mixed repeated models were run separately within non-pregnant and pregnant mice to assess the effects of diet and day, and separately within the SLD and HFHSD mice to assess the effects of pregnancy and day. Where diet × day and/or pregnancy × day interactions were significant, data for each day were analysed separately using a two-way ANOVA to assess the effects of pregnancy and diet. Diet effect, ^$P < 0.05$, ~$P < 0.01$, ~~$P < 0.001$. Pregnancy effect, *$P < 0.05$, **$P < 0.01$, ***$P < 0.001$. Within days, where a diet × pregnancy interaction was significant, one-way ANOVAs were used to assess the effects of diet within each pregnancy group, and also the effects of pregnancy within each diet group. NP-SLD *vs.* P-SLD-, $^{\$\$}P < 0.01$, $^{\$\$\$}P < 0.001$; NP-HFHSD *vs.* P-HFHSD, $^{@}P < 0.05$, $^{@@@}P < 0.001$; NP-SLD *vs.* NP-HFHSD, $^{\#}P < 0.05$, $^{\#\#}P < 0.01$, $^{\#\#\#}P < 0.001$; P-SLD *vs.* P-HFHSD, $^{+++}P < 0.001$. Exact *P*-values are reported in Table S2 on figshare (DOI: https://doi.org/10.25909/28079570).

interaction: $P = 0.003$) and pregnancy (pregnancy × diet interaction: $P = 0.003$), and the effects of pregnancy also changed across the study (day × pregnancy interaction: $P = 0.044$). We therefore analysed the effects of diet and pregnancy for each 2 day period throughout the study (Fig. 2*Bi*). Food intake was higher in SLD than HFHSD mice during acclimatisation, from Day 2.5 until Day 6.5 and again at Days 10.5–12.5 and 16.5–17.5 (all $P < 0.05$, Fig. 2*Bi*). Food intake from acclimatisation until Day 6.5 and at Days 10.5–12.5 and 16.5–17.5 did not differ between non-pregnant and pregnant groups (Fig. 2*Bi*). The effects of diet on food intake between Days 6.5–10.5 and 12.5–16.5 differed between pregnant and non-pregnant mice (pregnancy × diet interactions: all $P < 0.05$). On Days 6.5–8.5 and 8.5–10.5, food intake was not altered by pregnancy status within either diet group, or by diet within non-pregnant mice (Fig. 2*Bi*). Food intake during this period was higher in pregnant SLD-fed than HFHSD mice (both $P < 0.001$, Fig. 2*Bi*). On Days 12.5–14.5, food intake was not altered by pregnancy status, within either diet group, and was greater in SLD than HFHSD groups within both non-pregnant and pregnant groups (all $P < 0.05$, Fig. 2*Bi*). On Days 14.5–16.5, pregnant mice ate more than non-pregnant mice within the SLD groups ($P < 0.001$), but not within HFHSD groups (Fig. 2*Bi*). Food intake on Days 14.5–16.5 remained higher in SLD than HFHSD groups within both non-pregnant ($P = 0.015$) and pregnant groups ($P < 0.001$, Fig. 2*Bi*).

During the light-phase, the effects of pregnancy and diet on food intake varied with study day (day × pregnancy interaction: $P < 0.001$; day × diet interaction: $P = 0.043$, Fig. 2*Bii*). Within each study day, SLD mice ate more than HFHSD mice (all $P < 0.05$, Fig. 2*Bii*). From Day 8.5 onwards, food intake was higher in pregnant than non-pregnant mice (all $P < 0.05$, Fig. 2*Bii*).

During the dark-phase, the effects of pregnancy on food intake differed between diets (pregnancy × diet interaction: $P = 0.009$) and did not differ between study days (Fig. 2*Biii*). Food intake was higher in SLD than HFHSD mice within pregnant mice ($P < 0.001$) but did not differ between diet groups within non-pregnant mice (Fig. 2*Biii*). Food intake was higher in non-pregnant than pregnant mice within the HFHSD groups ($P < 0.001$)

but conversely was higher in pregnant than non-pregnant mice with SLD groups ($P = 0.016$, Fig. 2*Biii*).

**Energy per meal.** Across a 24 h period, the amount of energy consumed per meal was higher in HFHSD than SLD mice ($P = 0.015$), was unaffected by pregnancy and did not change across study days (Fig. 2*Ci*).

During the light-phase, the effects of pregnancy and diet on energy intake per meal differed between days (day × pregnancy and day × diet interactions: each $P < 0.01$, Fig. 2*Cii*), and we therefore analysed the effects of diet and pregnancy for each 2 day period throughout the study. HFHSD mice consumed more energy per light-phase meal than SLD mice on Days 0.5–2.5, but this pattern reversed with time, such that SLD mice consumed more energy per meal than HFHSD mice on Days 8.5–10.5 and 14.5–16.5 (all $P < 0.05$, Fig. 2*Cii*). Pregnant mice consumed more energy per light-phase meal than non-pregnant mice on most days from Day 2.5 onwards (all $P < 0.05$, Fig. 2*Cii*).

During the dark-phase, the effects of pregnancy and diet on energy intake per meal differed between days (day × pregnancy × diet interaction: $P = 0.023$, Fig. 2*Ciii*). During acclimatisation, on Days 0.5–2.5 and 6.5–8.5, energy intake per dark-phase meal was higher in HFHSD mice than SLD mice (each $P < 0.05$, Fig. 2*Ciii*). Pregnancy did not affect energy intake per meal during the dark-phase independent of diet at any day, but had diet-dependent effects on Days 14.5–16.5 (pregnancy × diet interaction: $P = 0.049$, Fig. 2*Ciii*). On Days 14.5–16.5, energy intake per dark-phase meal was higher in pregnant compared to non-pregnant SLD mice ($P = 0.004$), but not HFHSD mice, and did not differ between diet groups within either pregnant or non-pregnant groups (Fig. 2*Ciii*).

**Meal size.** Across 24 h, the effect of diet on meal size (g) differed with pregnancy and day (day × pregnancy × diet interaction: $P = 0.007$, Fig. 2*Di*), and we therefore analysed the effects of diet and pregnancy on each day of the study. Meal size was higher in the SLD than HFHSD group during acclimatisation and on Days 0.5–2.5, 2.5–4.5, 4.5–6.5, 8.5–10.5, 10.5–12.5 and 16.5–17.5 (all

$P < 0.05$, Fig. 2*Di*). On study Days 6.5–8.5, 12.5–14.5 and 14.5–16.5, the effects of diet on meal size differed with pregnancy status (pregnancy × diet interactions: all $P < 0.05$, Fig. 2*Di*). On Days 6.5–8.5 and 12.5–14.5, SLD mice consumed larger meals than HFHSD mice within pregnant groups (both $P < 0.001$), but meal size was similar between diet groups within non-pregnant mice, and similar between non-pregnant and pregnant groups within each diet group (Fig. 2*Di*). On Days 14.5–16.5, SLD mice consumed larger meals than HFHSD mice within pregnant and non-pregnant groups (both $P < 0.01$). Meal size on Days 14.5–16.5 was greater in pregnant than non-pregnant mice within SLD groups ($P < 0.001$), but did not differ with pregnancy within HFHSD mice (Fig. 2*Di*).

During the light-phase, the effects of diet on meal size (g) differed with both day (day × diet interaction: $P < 0.001$) and pregnancy (pregnancy × diet interaction: $P = 0.013$), and the effects of pregnancy also changed across the study (day × pregnancy interaction: $P < 0.001$). We therefore analysed the effects of diet and pregnancy for each 2 day period throughout the study (Fig. 2*Dii*). SLD mice ate larger light-phase meals than HFHSD mice on most study days (Fig. 2*Dii*). Pregnant mice consumed larger light-phase meals than non-pregnant mice from Day 4.5 until Day 10.5 and again on Days 16.5–17.5 (pregnancy effects, each $P < 0.05$, Fig. 2*Dii*). On Days 2.5–4.5 and 14.5–16.5, the effects of pregnancy on meal size differed between diet groups (pregnancy × diet interactions: both $P < 0.05$, Fig. 2*Dii*). On Days 2.5–4.5, pregnant mice ate larger meals than non-pregnant mice in SLD mice ($P = 0.014$), but not in HFHSD mice (Fig. 2*Dii*). In addition, on Days 2.5–4.5, SLD mice ate larger meals than HFHSD mice in non-pregnant groups ($P = 0.023$), but not within pregnant groups (Fig. 2*Dii*). On Days 14.5–16.5, pregnant mice ate larger meals than non-pregnant mice in both SLD ($P < 0.001$) and HFHSD groups ($P = 0.013$). In addition, on Days 14.5–16.5, SLD mice ate larger meals than HFHSD mice within both non-pregnant ($P = 0.001$) and pregnant groups ($P < 0.001$, Fig. 2*Dii*).

During the dark-phase, the effect of diet on meal size (g) differed with pregnancy and day (day × pregnancy × diet interaction: $P = 0.007$, Fig. 2*Diii*), and we therefore analysed the effects of diet and pregnancy on each day of the study. For all study days except Days 14.5–16.5, SLD mice consumed larger dark-phase meals than HFHSD mice (all $P < 0.05$), and meal size did not differ between pregnant and non-pregnant mice (Fig. 2*Diii*). The effects of pregnancy on dark-phase meal size differed between diet groups on Days 14.5–16.5 (pregnancy × diet interaction: $P = 0.010$). On Days 14.5–16.5, pregnant mice ate larger dark-phase meals than non-pregnant mice in SLD groups ($P = 0.004$), but not in those fed HFHSD, while SLD mice ate larger meals than HFHSD mice within

both non-pregnant ($P = 0.028$) and pregnant groups ($P < 0.001$, Fig. 2*Diii*).

**Meal duration.** Across 24 h, meal duration was shorter in pregnant than non-pregnant mice ($P = 0.005$), and in HFHSD than SLD mice ($P < 0.001$), and decreased across time ($P < 0.001$), with no interactions between factors (Fig. 2*Ei*).

During the light-phase, meal duration was similarly shorter in HFHSD than SLD mice ($P < 0.001$) but did not differ between pregnant and non-pregnant mice or with study day (Fig. 2*Eii*). During the dark-phase, meal duration did not change with study day, and the effects of diet differed between pregnant and non-pregnant groups (diet × pregnancy: $P = 0.017$, Fig. 2*Eiii*). Within non-pregnant and pregnant groups, dark-phase meal duration was shorter in HFHSD than SLD mice (both $P < 0.001$, Fig. 2*Eiii*). Within SLD ($P < 0.001$) and HFHSD ($P = 0.004$) groups, dark-phase meal duration was shorter in pregnant than non-pregnant mice.

**Meal number.** Across 24 h, the number of meals consumed each day changed between study days ($P = 0.034$), being greater on Days 0.5–2.5 compared to acclimatisation for all mice ($P = 0.016$) and not different between other days (Fig. 2*Fi*).

During the light-phase, the effects of diet differed between study days (diet × day interaction: $P = 0.024$, Fig. 2*Fii*). During acclimatisation, SLD mice ate more light-phase meals than HFHSD mice ($P = 0.007$) with no differences between mice subsequently mated and non-mated (Fig. 2*Fii*). On Days 14.5–16.5, the effects of diet differed between pregnant and non-pregnant mice (diet × pregnancy interaction: $P = 0.012$). HFHSD mice ate more light-phase meals than SLD mice within pregnant ($P = 0.044$) but not non-pregnant groups. Pregnant mice ate more light-phase meals than non-pregnant mice on Days 14.5–16.5 within the HFHSD groups ($P = 0.006$), but not in those fed an SLD. Light-phase meal number was unaffected by diet or pregnancy on other study days (Fig. 2*Fii*).

During the dark-phase, the effects of pregnancy and diet on dark-phase meal number differed between days (day × pregnancy × diet interaction: $P = 0.028$, Fig. 2*Fiii*), and we therefore analysed the effects of diet and pregnancy on each day of the study. No main effects of pregnancy were observed for any period of the study. On Days 2.5–6.5, 12.5–14.5 and 16.5–17.5, the HFHSD group consumed more dark-phase meals than the SLD group (each $P < 0.05$, Fig. 2*Fiii*). On Days 0.5–2.5 and 8.5–10.5, the effect of pregnancy differed between diets (pregnancy × diet interaction: each $P < 0.05$). On Days 0.5–2.5, dark-phase meal number did not differ between pregnant and non-pregnant mice within SLD groups

but was greater in NP-HFHSD than P-HFHSD mice ($P = 0.001$, Fig. 2*Fiii*). Within non-pregnant mice, those fed an HFHSD ate more meals during the dark-phase ($P < 0.001$), but meal number did not differ between diet groups in pregnant mice (Fig. 2*Fiii*). On Days 8.5–10.5, dark-phase meal number did not differ between non-pregnant or pregnant groups within mice fed either SLD or HFHSD. Non-pregnant mice fed an HFHSD ate more meals during the dark-phase than mice fed an SLD ($P = 0.013$). Pregnancy had no effect on meal number irrespective of maternal diet (Fig. 2*Fiii*).

**Food intake and meal size at the end of the study.** In addition to investigating how feeding behaviours changed across the study, we analysed daily food intake (Fig. 3*Ai*, *ii* and *iii*) and average meal size in grams (Fig. 3*Bi*, *ii* and *iii*) during the final two study days separately, as GVA function is expected to correlate with current feeding behaviours and satiation. Across 24 h, the effects of diet on food intake and meal size differed between pregnancy groups (pregnancy × diet interactions, $P = 0.007$ and $P = 0.003$ respectively, Fig. 3*A* and *B*). We therefore separately analysed the effects of diet within each pregnancy group and the effects of pregnancy within each diet group for each outcome. Food intake was higher in pregnant SLD than HFHSD mice ($P < 0.001$) Fig. 3*A*. Food intake was also higher in pregnant (compared to non-pregnant) SLD mice ($P = 0.003$) but this was not the case in the HFHSD groups ($P = 0.448$; Fig. 3*A*). Meal size was larger in SLD than HFHSD mice within both pregnant ($P < 0.001$) and non-pregnant groups ($P = 0.029$, Fig. 3*B*). Meal size was also higher in pregnant than non-pregnant mice within SLD mice ($P = 0.002$) but not HFHSD mice (Fig. 3*B*).

During the light-phase, food intake was higher in SLD than HFHSD mice (diet: $P < 0.001$) and in pregnant than non-pregnant mice (pregnancy: $P < 0.001$) (Fig. 3*Aii*). Similarly, light-phase meal size was larger in SLD than HFHSD mice (diet: $P < 0.001$) and in pregnant than non-pregnant mice (pregnancy: $P < 0.001$), with no interactions (Fig. 3*Bii*).

During the dark-phase, the effects of diet on food intake differed with pregnancy status (pregnancy × diet

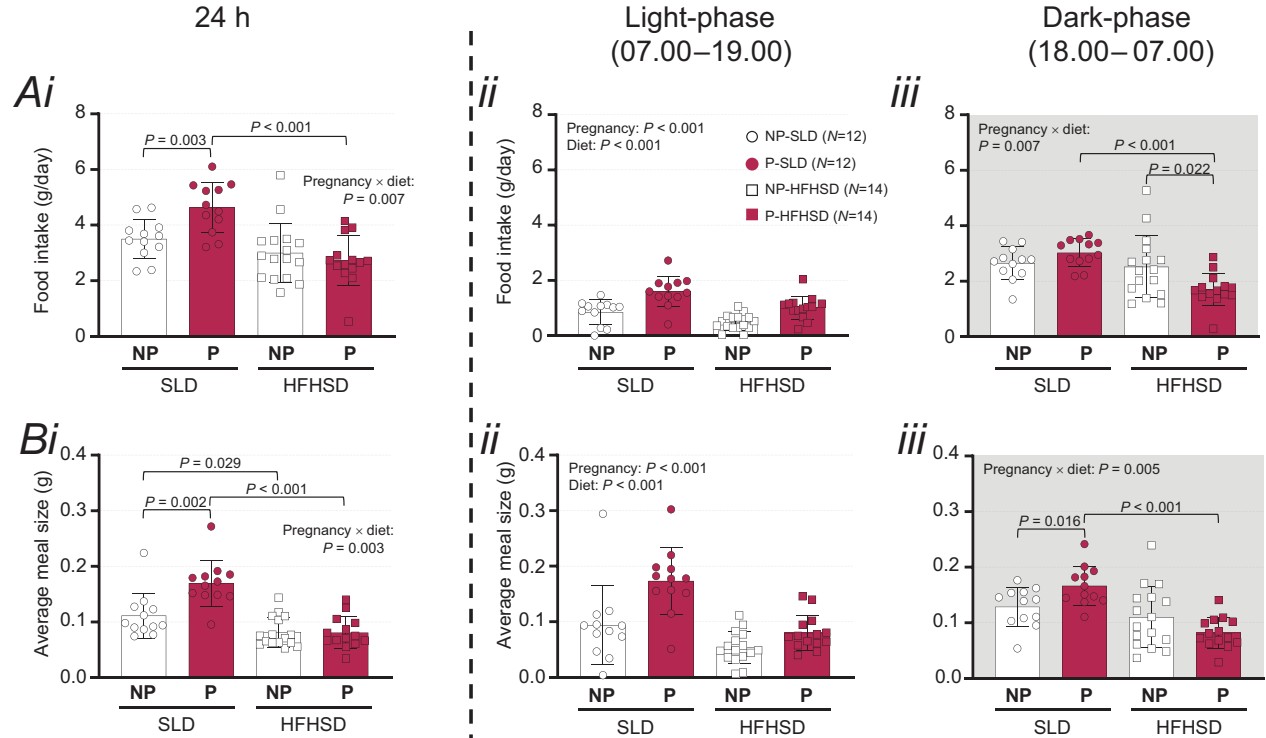

**Figure 3. Impact of diet and pregnancy on total food intake and average meal size during the last 2 days of the study (Days 16.5–17.5)**
Total food (g, *Ai*, *ii* and *iii*) and average meal size (g, *Bi*, *ii* and *iii*) of pregnant (P) and non-pregnant (NP) mice exposed to a standard laboratory diet (SLD, non-pregnant, NP, open circles: $N = 12$; pregnant, P, closed circles: $N = 12$) and high-fat high-sugar diet (HFHSD, NP, open squares: $N = 16$; P, closed squares: $N = 14$). Data are presented separately for the entire 24 h, light-phase and dark-phase (shaded). Bars and whiskers show mean ± SD, with data from each mouse indicated by symbols. Data were analysed using a two-way ANOVA. Where diet × pregnancy interactions were significant, one-way ANOVAs were used to assess the effects of pregnancy within each diet group, and the effects of diet within non-pregnant and pregnant groups. NP-SLD *vs.* P-SLD, $^{\$}P < 0.05$, $^{\$\$}P < 0.01$; NP-HFHSD *vs.* P-HFHSD, $^{@}P < 0.05$; P-SLD *vs.* P-HFHSD, $^{+++}P < 0.001$.

interaction: $P = 0.007$, Fig. 3*Aiii*). Dark-phase food intake was higher in SLD than HFHSD mice within pregnant ($P < 0.001$) but not non-pregnant groups ($P = 0.102$; Fig. 3*Aiii*). Dark-phase food intake was higher in non-pregnant than pregnant mice within HFHSD ($P = 0.022$) but not SLD groups (Fig. 3*Aiii*). During the dark phase, the effect of diet on meal size also differed with pregnancy status (pregnancy × diet interaction: $P = 0.005$, Fig. 3*Biii*). Dark-phase meal size was higher in SLD than HFHSD mice within pregnant ($P < 0.001$) but not non-pregnant groups ($P = 0.315$; Fig. 3*Biii*). Dark-phase food intake was higher in pregnant than non-pregnant mice within SLD mice ($P = 0.016$) but did not differ between pregnant and non-pregnant HFHSD groups ($P = 0.097$; Fig. 3*Biii*).

**Impacts of diet and pregnancy on the mechanosensitivity of GVAs and correlation with meal size.** The response of mucosal afferents to stroking increased as the von Frey hair weight increased ($P < 0.001$) but was unaffected by pregnancy or diet (both $P > 0.1$, Fig. 4*A*). Responses of GVA mucosal afferents to stroking (200 mg) were not correlated with light-phase meal size within either diet group (Fig. 4*C*).

The effects of pregnancy and diet on tension-sensitive GVA mechanosensitivity differed with load (load × pregnancy × diet interaction: $P = 0.003$, Fig. 4*B*). At loads of 0–2 g, tension-sensitive GVA responses were unaffected by diet or pregnancy. At greater loads, the effects of diet on tension-sensitive GVA responses differed between pregnancy groups (pregnancy × diet interactions: each $P < 0.05$, Fig. 4*B*). Within non-pregnant groups, tension-sensitive GVA responses to loads of 3 g ($P = 0.039$), 4 g ($P = 0.009$) and 5 g tension ($P = 0.026$) were greater in SLD than HFHSD mice. In contrast, tension-sensitive GVA responses did not differ between diet groups within pregnant mice (Fig. 4*B*). Within SLD mice, tension-sensitive GVA responses were greater in non-pregnant than pregnant mice at loads of 3 g ($P = 0.048$), 4 g ($P = 0.030$) and 5 g ($P = 0.022$). In contrast, within HFHSD mice, tension-sensitive GVA responses did not differ between non-pregnant and pregnant mice (Fig. 4*B*). Responses of tension-sensitive GVA to a 5 g load were not correlated with light-phase meal size within either diet group (Fig. 4*D*). Representative recordings of tension-sensitive GVA responses in each group at 3 g load are shown in Fig. 4*E–H*.

### Impact of diet and pregnancy on the transcript expression of receptors associated with GVA modulation

Expression of *CCKA*, *GHSR*, *LepR* and *TRPV1* (Fig. 5*A–D*) was unaffected by diet ($P = 0.201$, $P = 0.665$,

$P = 0.914$ and $P = 0.142$, respectively) or pregnancy ($P = 0.569$, $P = 0.316$, $P = 0.970$ and $P = 0.609$, respectively). Relative abundance of *GHR* was higher in pregnant compared to non-pregnant mice (Fig. 5*E*, $P = 0.005$) but was unaffected by diet ($P = 0.172$).

## Discussion

The current study investigated the effects of an HFHSD and pregnancy on food intake and GVA signalling. The HFHSD mice consumed more energy across a 24 h period than the SLD mice up until Day 8.5, due to greater dark-phase energy intake. Daily food intake (g) was higher in P-SLD than P-HFHSD mice from Days 6.5 to 16.5, predominantly due to greater food intake during the dark-phase. From mid-pregnancy onwards, pregnant mice consumed more food (g and kJ) than non-pregnant mice during the light-phase due to greater meal size, independent of diet. We also demonstrated that the response of GVAs to stretch was attenuated by pregnancy within mice fed an SLD, and by HFHSD-feeding within non-pregnant mice, but that there was no further reduction in GVA mechanosensitivity during pregnancy within mice fed an HFHSD. A previous report demonstrated that GVA responses to stretch correlate with meal size in the setting of pregnancy (Li et al., 2021), with greater meal size in the last 2 days of the current study in pregnant compared to non-pregnant SLD mice. Since tension-sensitive GVA mechanosensitivity is reduced by HFHSD-feeding in non-pregnant mice, which would be expected to permit larger meals, their smaller meal sizes relative to NP-SLD mice probably involve other satiety mechanisms.

### Greater food intake during pregnancy occurs concurrent with downregulation of tension-sensitive GVAs in SLD mice

Maternal body weight increases during pregnancy in humans and rodents (Clarke et al., 2021), and consistent with this, pregnant mice gained more weight than non-pregnant mice in both diet groups. Within the SLD group, mice ate more food (g) later in pregnancy (Days 14.5–16.5) than non-pregnant mice, predominantly due to an increase in light-phase meal size. This is consistent with our previous findings in SLD mice (Li et al., 2021) and occurred without adaptions in meal number (Ladyman et al., 2018; Li et al., 2021). Pregnant SLD mice also had a shorter meal duration across the dark-phase compared to NP-SLD mice, as previously reported (Li et al., 2021). On the final 2 days of the study (Days 16.5–17.5), total food intake and meal size (g) were greater in pregnant compared to NP-SLD mice. These increases are consistent with downregulation of satiety signals during

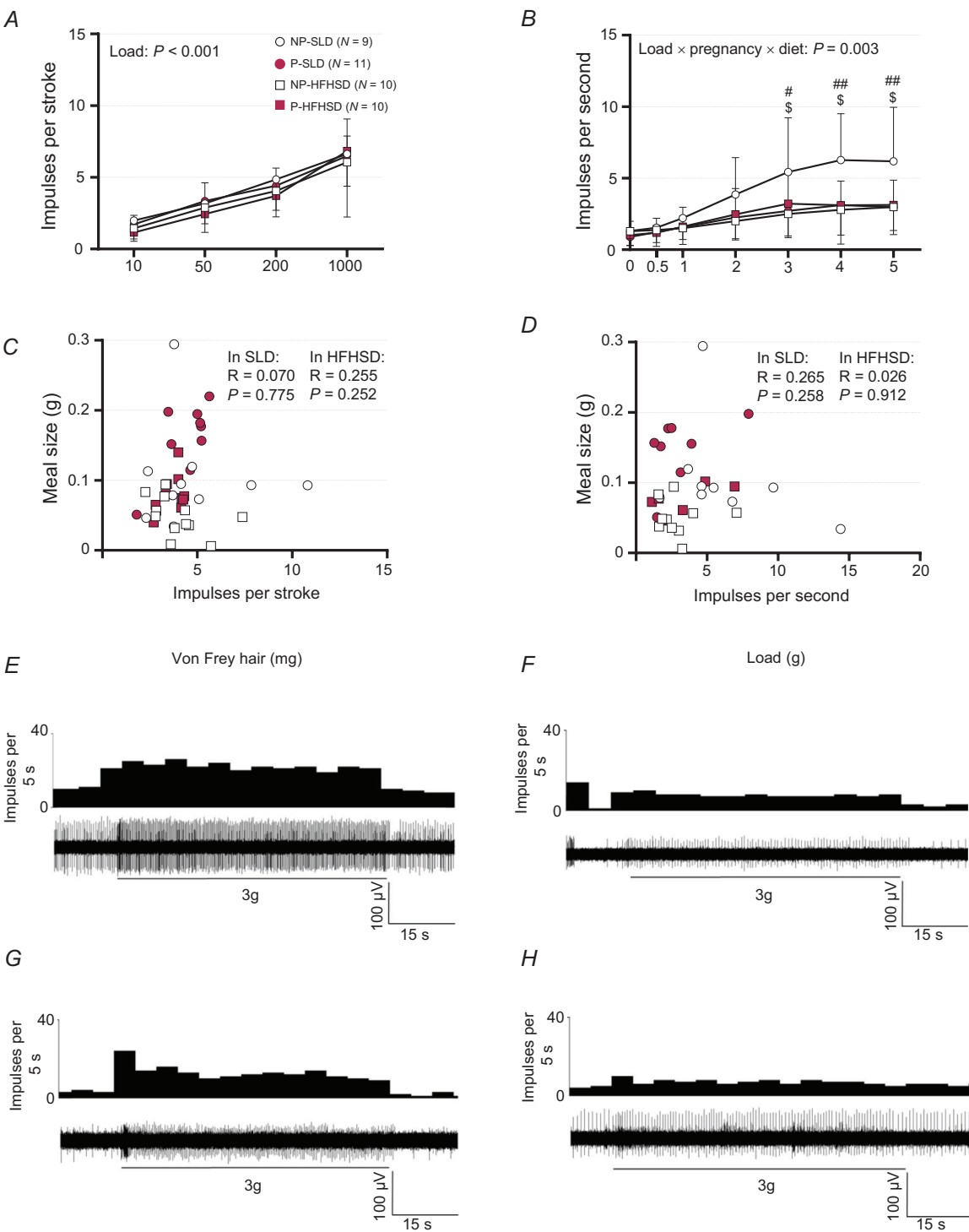

**Figure 4. Impact of diet and pregnancy on gastric vagal afferent mechanosensitivity and correlation to meal size**

*A* and *B*, the response of mucosal gastric vagal afferents (GVAs) to mucosal stroking (10–1000 mg, *A*) and tension-sensitive GVAs to circular tension (0–5 g, *B*) in pregnant (P) and non-pregnant (NP) mice fed standard laboratory diet (SLD, non-pregnant, NP, open circles: *N* = 9; pregnant; P, closed circles: *N* = 11) or high-fat high-sugar diet (HFHSD, NP, open squares: *N* = 10; P, closed squares: *N* = 10). Data are presented as mean ± SD and were analysed using a linear mixed model to assess the effect of pregnancy (pregnant *vs.* non-pregnant) and diet (SLD *vs.* HFHSD) with load [circular tension (g) or von Frey hair (mg), respectively] as a repeated factor. Where a pregnancy × diet × load interaction was significant, mixed repeated models were used to assess the effects of diet

and day separately in non-pregnant and pregnant groups, and the effects of pregnancy and day separately within SLD and HFHSD mice. Where diet × load and/or pregnancy × load interactions were significant, a two-way ANOVA was run to assess the effects of pregnancy and diet separately for each level of load. Where a diet × pregnancy interaction was significant for a given load, we ran a one-way ANOVA to assess the effects of pregnancy within each diet group and of diet within each pregnancy group. NP-SLD *vs.* P-SLD, $^\$P < 0.05$; NP-SLD *vs.* NP-HFHSD, $^\#P < 0.05$, $^{\#\#}P < 0.01$. Exact *P*-values are reported in Table S3 on figshare (DOI: https://doi.org/10.25909/28079570). *C* and *D*, the correlation between the response of (*C*) mucosal afferents to mucosal stroking with a 200 mg von Frey hair and (*D*) tension-sensitive GVA to 5 g tension and light-phase meal size. Data were analysed using a Pearson's correlation. Symbols indicate outcomes for individual animals. *E–H*, typical response of a tension-sensitive GVA to 3 g load in NP-SLD (*E*), P-SLD (*F*), NP-HFHSD (*G*) and P-HFHSD (*H*) mice.

pregnancy, including the attenuated mechanosensitivity of tension-sensitive GVAs in P-SLD mice in our previous (Li et al., 2021) and the current study. Unlike our previous study (Li et al., 2021), there was no association between meal size and GVA responses to stretch in the present cohort. This is probably due to a smaller sample size, although the increased meal size during pregnancy may reflect downregulation of multiple satiety factors contributing to meal size, including intestinal satiety signals (Clarke, Li, et al., 2023; Cummings & Overduin, 2007; Ladyman et al., 2011) and central food intake regulatory pathways (e.g. leptin resistance, as reviewed in Clarke et al., 2021). Dampened tension-sensitive GVA mechanosensitivity during pregnancy is probably driven by changes in hormone levels (Clarke et al., 2021) or their signalling pathways. In the whole nodose ganglia, mRNA expression of receptors that regulate vagal afferent sensitivity, including *TRPV1* (Kentish, Frisby, et al., 2015), *CCKA* (Daly et al., 2011), *GSHR* (Kentish et al., 2012) and *LepR* (Kentish, O'Donnell, et al., 2013), were unaffected by pregnancy, suggesting they are not involved in the observed changes in GVA mechanosensitivity during pregnancy in SLD mice. In contrast, expression of *GHR* in the nodose ganglia was increased in pregnancy regardless of diet. We have previously reported that *ex vivo* administration of growth hormone (GH) decreases the response of murine tension-sensitive GVAs to stretch (Li et al., 2021), and that circulating GH increases from early to mid-pregnancy and then remains elevated during late pregnancy in mice (Gatford et al., 2017). Neuron-specific GHR ablation reduced food intake in pregnant mice, suggesting that at least part of pregnancy-induced hyperphagia is driven by central GHR signalling (Teixeira et al., 2019). Together, this suggests GH downregulates peripheral as well as central satiety pathways to permit increased food intake during pregnancy.

## Weight gain and food intake are increased by feeding an HFHSD in non-pregnant mice, concurrent with downregulation of tension-sensitive GVAs

In the current study, HFHSD mice gained more weight than SLD mice during the 12 weeks from weaning, but at slower rates than in previous studies (Park et al., 2020) and without reaching heavier weights 12 weeks after diet commencement. During the 17.5 days of pregnancy, HFHSD mice were heavier than SLD mice except for on one day (Day 15). Within the present study, Glu Venus mice were maintained on a C57BL/6 background, an established strain for diet-induced obesity (Wang & Liao, 2012). HFHS diets, consistent with those used in the current study, induce weight gain in rats (Chen et al., 2019) and mice (Park et al., 2020), from as early as 4 weeks of feeding in female C57BL/6J mice (Park et al., 2020). Differences in the timing and extent of weight gain between studies could reflect the age at which the mice started the diet, which was introduced soon after weaning at 3–4 weeks old in the current study compared to 7 weeks old in the previous study (Park et al., 2020).

In the current study, 24 h energy intake was ∼50% higher in the non-pregnant HFHSD than SLD mice on selected days, predominantly due to an increase in total energy intake and meal number during the dark phase. Consequently, diurnal rhythmicity in feeding patterns was preserved in both diet groups, with HFHSD and SLD mice consuming ∼75% and ∼70% of their diet respectively during the dark-phase. This contrasts with the effects of HFD-induced obesity, which dampens diurnal rhythms in food intake in mice, increasing food intake during the light-phase such that these mice only consumed ∼60% of their food during the dark phase, while controls ate ∼80% of their diet during the dark-phase (Christie et al., 2018; Kentish et al., 2016). Differences between studies may be due to the different diets used (HFD *vs.* HFHSD). In part, this may also reflect the difference in adiposity induced by these obesogenic diets, with a significant increase in adiposity in the HFD mice reported in the previous studies (>300% increase in gonadal fat pad mass) (Christie et al., 2018) and a smaller difference in adiposity between NP-HFHSD and NP-SLD mice in the current study (∼60% increase in gonadal fat pad mass). It is therefore possible the obese state rather than the diet induced the previously reported changes in diurnal food intake patterns. However, this study formed part of a larger cohort study where HFHSD feeding resulted in reduced insulin sensitivity, hyperinsulinaemia and impaired glucose tolerance (O'Hara et al., 2023), and therefore there are still diet-induced metabolic changes that may impact on circadian patterns of food intake patterns.

In the current study, a chronic HFHSD also dampened tension-sensitive GVA responses to stretch in non-pregnant mice. This is consistent with the reduced sensitivity of GVAs in female mice (Kentish et al., 2012) and intestinal VAs in male mice (Daly et al., 2011) fed an HFD [60% of energy from fat (lard)]. Despite the dampened tension-sensitive GVA signalling in HFHSD mice in the present study, they ate less food and smaller meals (g) during the light-phase in the 2 days prior to the electrophysiology experiments, compared to the SLD mice. Interestingly, the effects of diet on food intake were independent of pregnancy status, with lower food intake

in HFHSD than SLD mice despite this downregulation of GVA responses, suggesting that other satiety mechanisms are limiting overall food intake. For example, there are high levels of fat in the HFHSD and fat is a strong satiety mediator, with signals arising peripherally in the small intestine (Maher & Clegg, 2019). Fat-induced release of gut satiety hormones, such as cholecystokinin (CCK) and glucagon-like peptide 1 (GLP-1), can then either enter the bloodstream to bind to CCK and GLP1 receptors directly in the brain and/or act locally by binding to these receptors on VAs to reduce food intake (Clarke et al., 2021; Maher & Clegg, 2019). In HFD mice, responses

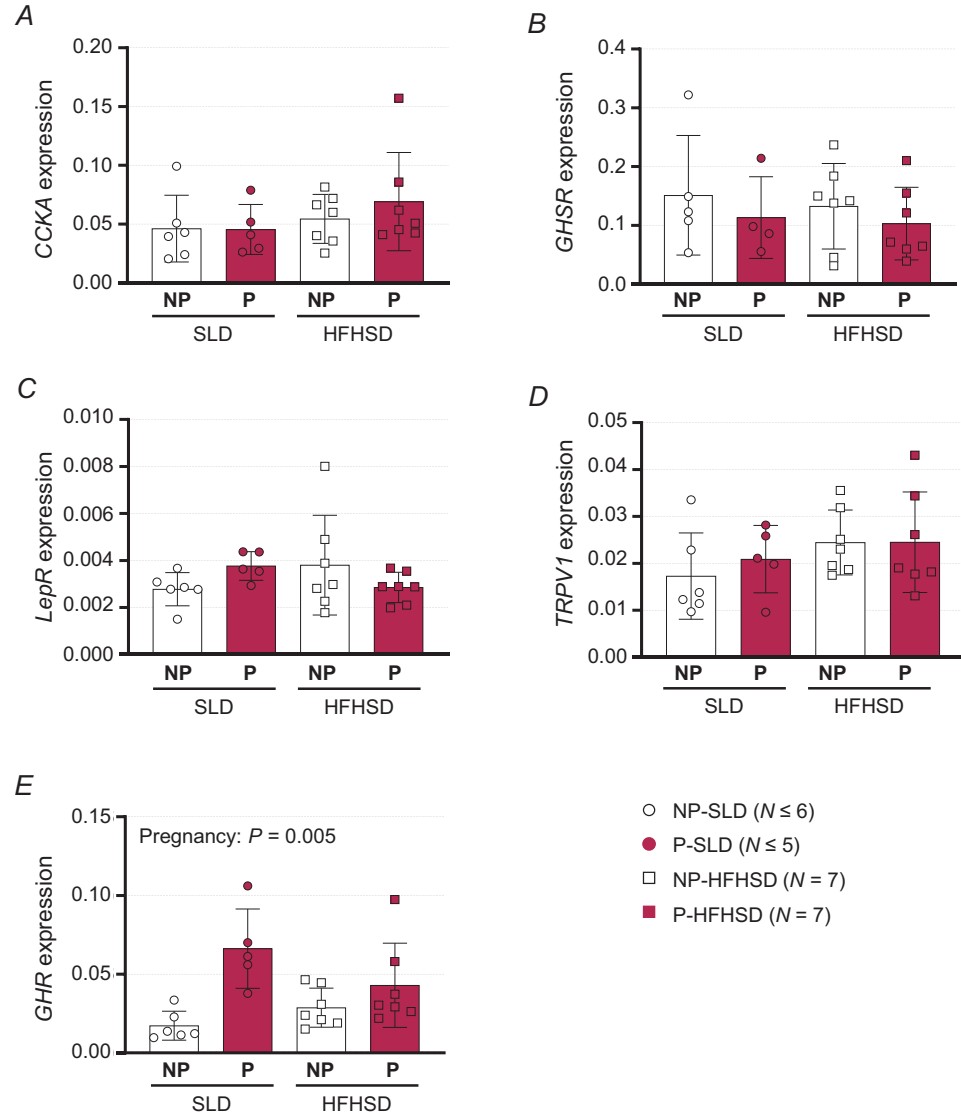

**Figure 5. Impact of diet and pregnancy on expression of receptors associated with GVA modulation**
*A–E*, relative mRNA expression of cholecystokinin receptor A (*A*, CCKA), ghrelin receptor (*B*, GSHR), leptin receptor (*C*, LepR), transient receptor potential vanilloid 1 (*D*, TRPV1) and growth hormone receptor (*E*, GHR) in pregnant (P) and non-pregnant (NP) mice fed standard laboratory diet (SLD, non-pregnant, NP, open circles: *N* ≤ 6; pregnant, P, closed circles: *N* ≤ 5) or high-fat high-sugar diet (HFHSD, NP, open squares: *N* = 7; P, closed squares: *N* = 7). Expression is relative to the average Ct of the reference genes *B2M* and *ACTB*. Bars show mean ± SD, and symbols show data for individual mice.

of jejunal VAs to CCK are dampened compared to responses in SLD mice (Daly et al., 2011) but this may not be the case with the HFHSD mice. Alternatively, given the reduced insulin sensitivity, hyperinsulinaemia and impaired glucose tolerance of pregnant HFHSD mice (O'Hara et al., 2023) it is possible that elevated post-prandial glucose might also contribute to reduced GVA mechanosensitivity. Similar patterns of reduced insulin sensitivity and impaired glucose tolerance are also seen during non-obese mouse and human pregnancy (Lain & Catalano, 2007; Musial et al., 2016; Powe et al., 2019). However, whether elevated glucose contributes in pregnancy and HFHSD-fed animals to suppress GVA mechanosensitivity requires further investigation since the impact of elevated glucose on GVA mechanosensitivity has not been reported.

## In HFHSD mice there is no further downregulation of GVA responses during pregnancy, but food intake is still higher in pregnant than non-pregnant HFHSD mice

Within the current study, maternal weight gain, litter size, average pup weight and perirenal fat pad weight of pregnant mice were unaffected by diet, while gonadal fat mass was heavier in HFHSD than SLD mice regardless of pregnancy status. This contrasts with findings from Park et al. (2020) where HFHSD-fed mice were heavier than SLD-fed mice at Days 7, 14 and 20 of pregnancy, with greater fat mass and lower lean mass as measured by echo magnetic resonance imaging in late pregnancy (Day 16). The greater weight gain reported by Park et al. (2020) could reflect average litter size/weight or more acute effects of an HFHSD on food intake, since female mice were on the diet for 6 weeks before mating compared to 12 weeks in the current study.

There were no differences in 24 h energy intake between P-HFHSD and NP-HFHSD mice in the current study across the 17.5 days of pregnancy. In addition, there was no further reduction in tension-sensitive GVA mechanosensitivity during pregnancy in the HFHSD mice. This is consistent with no effect of pregnancy on meal size during the light-phase in the HFHSD groups, 2 days prior to the electrophysiology recordings. However, food intake during the dark-phase in the final two study days was significantly reduced in P-HFHSD compared to NP-HFHSD mice, which may reflect other pregnancy-related adaptations. GVAs exhibit circadian rhythmicity, with the mechanosensitivity of tension-sensitive GVAs exhibiting the greatest responsiveness during the light-phase, aligning with lower energy demands and reduced food intake (Kentish, Frisby, et al., 2013). These rhythms are lost in HFD-induced obese mice, due to attenuated mechanosensitivity of

tension-sensitive GVAs during the light-phase compared to SLD controls (Kentish et al., 2016). Recently, we have shown an increase in light-phase food intake during pregnancy, predominantly due to an increase in food intake bouts occurring late in the light-phase (Zeitgeber time 8–12) (Clarke, Vincent, et al., 2023). Since tissue was collected early in the light-phase for GVA recordings in the current study, it is possible that adaptations in P-HFHSD mice were not captured, and therefore future research should characterise daily variation in GVA sensitivity in response to an HFHSD and pregnancy. Although nodose ganglia GHR expression was upregulated during pregnancy regardless of diet, GHR-mediated down-regulation of GVA responses to stretch may be impaired in obesity due to lower circulating GH abundance. In women, circulating concentrations of the placental GH variant are negatively correlated with maternal body mass index during pregnancy (Coutant et al., 2001; Verhaeghe et al., 2002). Whether obesity similarly down-regulates circulating GH abundance in species where GH remains pulsatile during pregnancy and is not secreted by the placenta, like mice, has not been evaluated. Nevertheless, obesity downregulates circulating GH in non-pregnant mice, as in non-pregnant humans (Steyn et al., 2013). Lastly, the stage of pregnancy at which pregnancy adaptations in GVA mechanosensitivity are lost is unknown. We therefore suggest that future research should measure GVA sensitivity across early, mid- and late pregnancy in HFHSD mice, similar to our prior study using SLD mice (Li et al., 2021).

## Responses of mucosal GVAs are not changed during pregnancy or by diet

The response of mucosal afferents to mucosal stroking was unchanged during pregnancy or by an HFHSD, consistent with our prior findings in pregnant compared to NP-SLD mice (Li et al., 2021) and the lack of difference in mucosal GVA responses between female HFD- and SLD-fed mice (Kentish et al., 2012). Mucosal afferents are located within the gastric mucosa and are thought to modulate gastric emptying through discrimination of particle size (Page et al., 2002). Reported changes in rates of gastric emptying during pregnancy are inconsistent. Studies have reported no change in humans (Chiloiro et al., 2001; Macfie et al., 1991; O'Sullivan, 1993; Whitehead et al., 1993) or slower gastric emptying in both humans (Levy et al., 1994) and rats (Shah et al., 2000). In relation to obesity, similar controversies exist, with no difference in gastric emptying rates between obese and lean individuals (Buchholz et al., 2013; Verdich et al., 2000), and increased (Baudry et al., 2012) or decreased (Kentish, Ratcliff, et al., 2015) gastric emptying in obese compared to lean mice. Even less is understood about the dual effects of obesity and

pregnancy; in one study, gastric emptying of water was similar in obese compared to lean late-pregnant women (Wong et al., 2007). Overall, if mucosal afferents play a major role in gastric emptying, it appears that pregnancy and obesity may not alter this response, but this requires further investigation.

## Conclusion

An HFHSD reduces tension-sensitive GVA responses to stretch in non-pregnant mice, similar to observations in HFD-induced obesity. However, meal size (g) is reduced rather than increased in HFHSD mice, suggesting other satiety mechanisms are contributing to food intake behaviour. This study also confirmed previous findings that mechanosensitivity of tension-sensitive GVAs is selectively attenuated during a lean murine pregnancy, with concurrent increases in total food intake and meal size (Li et al., 2021). Although there is a reduction in the mechanosensitivity of tension-sensitive GVAs in HFHSD mice compared to SLD mice, there was no further reduction in mechanosensitivity of tension-sensitive GVAs in P-HFHSD compared to NP-HFHSD mice. Further studies are required to increase understanding of food intake regulation across pregnancy to inform strategies to improve pregnancy outcomes.

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

## Additional information

### Data availability statement

Data will be made available upon request.

### Competing interests

None declared.

### Author contributions

L.M.N. established and provided access to the experimental model of HFHSD feeding during pregnancy in the Glu Venus mice. All authors contributed to the experimental design. G.C., H.L., E.H., K.L.G. and A.J.P. conducted experiments. G.C. analysed the data and wrote the manuscript. All authors contributed to interpretation of the data and editing of the manuscript.

### Funding

G.S.C. was supported by an Australian Government Research Training Program (RTP) Stipend Scholarship. L.M.N. was supported by a C. J. Martin Fellowship from the National Health and Medical Research Council (GNT1092158). This research was partly funded by the University of Adelaide, Faculty of Health and Medical Sciences Strategic Grant (2022).

### Acknowledgements

We thank Professors Frank Reimann and Fiona Gribble from the Wellcome-MRC Institute of Metabolic Science-Metabolic Research Laboratories (Cambridge, UK) for provision of the genetic mouse line. We thank the SAHMRI bioresources facility and animal technicians.

Open access publishing facilitated by The University of Adelaide, as part of the Wiley - The University of Adelaide agreement via the Council of Australian University Librarians.

## Keywords

food intake, high-fat high-sugar diet, mouse, pregnancy, vagal afferents

## Supporting information

Additional supporting information can be found online in the Supporting Information section at the end of the HTML view of the article. Supporting information files available:

**Peer Review History**

