## [Peer Review History · The Journal of Physiology]

Pregnancy and a high-fat high-sugar diet each attenuate mechanosensitivity of murine gastric vagal afferents, with no additive effects.

Georgia S Clarke, Hui Li, Elaheh Heshmati, Lisa Marie Nicholas, Kathryn L Gatford, and Amanda J Page
DOI: 10.1113/JP286115

Corresponding author(s): Amanda Page (amanda.page@adelaide.edu.au)

Review Timeline:

Submission Date:	10-Dec-2023
Editorial Decision:	27-Feb-2024
Revision Received:	22-Dec-2024
Editorial Decision:	16-Jan-2025
Revision Received:	21-Jan-2025
Accepted:	03-Feb-2025

Senior Editor: Kim Barrett

Reviewing Editor: Kim Barrett

Transaction Report:

Dear Dr Page,

Re: JP-RP-2023-286115 "Pregnancy and a high-fat high-sugar diet each attenuate mechanosensitivity of murine gastric vagal afferents, with no additive effects." by Georgia S Clarke, Hui Li, Lisa M Nicholas, Kathryn L Gatford, and Amanda J Page

Thank you for submitting your manuscript to The Journal of Physiology. It has been assessed by a Reviewing Editor and by 1 expert referee and we are pleased to tell you that it is potentially acceptable for publication following satisfactory major revision.

LANGUAGE EDITING AND SUPPORT FOR PUBLICATION: If you would like help with English language editing, or other article preparation support, Wiley Editing Services offers expert help, including English Language Editing, as well as translation, manuscript formatting, and figure formatting at www.wileyauthors.com/eoo/preparation. You can also find resources for Preparing Your Article for general guidance about writing and preparing your manuscript at www.wileyauthors.com/eoo/prepresources.

REVISION CHECKLIST:

Please upload two versions of your manuscript text: one with all relevant changes highlighted and one clean version with no changes tracked. The manuscript file should include all tables and figure legends, but each figure/graph should be uploaded as separate, high-resolution files. The journal is now integrated with Wiley's Image Checking service. For further details, see: <https://www.wiley.com/en-us/network/publishing/research-publishing/trending-stories/upholding-image-integrity-wileys->

image-screening-service

We look forward to receiving your revised submission.

Yours sincerely,

Kim Barrett
Senior Editor
The Journal of Physiology

REQUIRED ITEMS

- Author photo and profile. First or joint first authors are asked to provide a short biography (no more than 100 words for one author or 150 words in total for joint first authors) and a portrait photograph. These should be uploaded and clearly labelled together in a Word document with the revised version of the manuscript. See Information for Authors for further details.
- You must start the Methods section with a paragraph headed Ethical Approval. A detailed explanation of journal policy and regulations on animal experimentation is given in Principles and standards for reporting animal experiments in The Journal of Physiology and Experimental Physiology by David Grundy J Physiol, 593: 2547-2549. doi:10.1113/JP270818). A checklist outlining these requirements and detailing the information that must be provided in the paper can be found at: <https://physoc.onlinelibrary.wiley.com/hub/animal-experiments>. Authors should confirm in their Methods section that their experiments were carried out according to the guidelines laid down by their institution's animal welfare committee, and conform to the principles and regulations as described in the Editorial by Grundy (2015), including an ethics approval reference number. The Methods section must contain a statement about access to food, water and housing, details of the anaesthetic regime: anaesthetic used, dose and route of administration, and method of killing the experimental animals.
- Please upload separate high-quality figure files via the submission form.
- Papers must comply with the Statistics Policy: https://jp.msubmit.net/cgi-bin/main.plex?form_type=display_requirements#statistics.

In summary:

- If $n \leq 30$, all data points must be plotted in the figure in a way that reveals their range and distribution. A bar graph with data points overlaid, a box and whisker plot or a violin plot (preferably with data points included) are acceptable formats.
- If $n > 30$, then the entire raw dataset must be made available either as supporting information, or hosted on a not-for-profit repository, e.g. FigShare, with access details provided in the manuscript.
- 'n' clearly defined (e.g. x cells from y slices in z animals) in the Methods. Authors should be mindful of pseudoreplication.
- All relevant 'n' values must be clearly stated in the main text, figures and tables.
- The most appropriate summary statistic (e.g. mean or median and standard deviation) must be used. Standard Error of the Mean (SEM) alone is not permitted.
- Exact p values must be stated. Authors must not use 'greater than' or 'less than'. Exact p values must be stated to three significant figures even when 'no statistical significance' is claimed.

- Please include an Abstract Figure file, as well as the Figure Legend text within the main article file. The Abstract Figure is a piece of artwork designed to give readers an immediate understanding of the research and should summarise the main conclusions. If possible, the image should be easily 'readable' from left to right or top to bottom. It should show the physiological relevance of the manuscript so readers can assess the importance and content of its findings. Abstract Figures should not merely recapitulate other figures in the manuscript. Please try to keep the diagram as simple as possible and without superfluous information that may distract from the main conclusion(s). Abstract Figures must be provided by authors no later than the revised manuscript stage and should be uploaded as a separate file during online submission labelled as File Type 'Abstract Figure'. Please also ensure that you include the figure legend in the main article file. All Abstract Figures should be created using BioRender. Authors should use The Journal's premium BioRender account to export high-resolution images. Details on how to use and access the premium account are included as part of this email.

EDITOR COMMENTS

Reviewing Editor:

This is an interesting study by a group leader in the field. The study is well designed and well written. It demonstrates that GVA mechanosensitivity are altered by obesogenic diet during pregnancy does not alter changes induced by pregnancy observed in control mice. The effects of interactions between diet and obesity are novel. However, in its current form the study lacks mechanistic insights and remains largely descriptive.

Major concerns:

1) It is not clear for the reviewer why was GVA sensitivity studied solely on one time point ie, almost at the end of the dynamic process of pregnancy. Additional time points, for studying GVA sensitivity should be provided. In particular based on the metabolic study, the period d 6,5 to 8,5 appears also to be of interest. Indeed, at this time point, both meal size as well as food intake are different between groups of interests. IN addition, at this time point body weight also appears different between P SLD and P HFHSD suggesting that changes in GVA sensitivity at this time could contribute to the increased weight gain of PSLD mice.

2) From a mechanistical point of view, more data should be provided describing the interactions between diet and pregnancy upon the expression of known or candidate receptors or factors involved GVA mechanosensing during pregnancy and/or obesity. For instance, the expression of TRPV1, GLP1R or pregnancy associated hormones should be described to support the absence of difference.

3) In fig 1, it is not clear why there is no difference in body weight gain is observed between P SLD and P HFHSD as at the beginning of mating HFHSD mice have a greater body weight than SLD mice?

Senior Editor:

The editor apologizes sincerely for the significant delays that occurred during the review of this manuscript. This resulted from challenges in identifying willing reviewers. The Reviewing Editor conducted an independent review and indicates that the manuscript, while interesting, lacks mechanistic information. The external reviewer actually suggests some additional mechanistic studies that could be performed. In light of the delays that have occurred, I would like to offer you the opportunity to revise your manuscript if additional data can be generated. Alternatively, we can offer to refer your manuscript to Experimental Physiology. No matter what, thank you for submitting your work to JP.

Precise p values must be supplied for all statistical comparisons.

Please also see 'Required Items' above.

REFEREE COMMENTS

Referee #1:

In the study titled "Pregnancy and a high-fat high-sugar diet each attenuate mechanosensitivity of murine gastric vagal

afferents, with no additive effects," Georgia Clarke and her collaborators explored the capacity of gastric vagal afferents (GVA) to modulate their mechanosensitive response under two physiological conditions: gestation and exposure to a high-fat high-sugar diet (HFHSD), both individually and in combination.

The observational results are well-executed in metabolic cage and vagal nerve electrophysiology during stretching. While the results are interesting and support previous findings, they seem overly observational and focus too much on stretching, a crucial aspect of satiety, but neglect nutrient sensing especially when dealing with HFHSD, and potential obese state.

Indeed, vagal afferent response to stretch correlates with meal size (Li et al., 2021), as validated in this manuscript. However, as suggested by the authors, vagal response is not solely dependent on stretching; there is a nutrient-dependent response, especially to fats and sugars activating gastric vagal afferents.

Major points:

Prolonged exposure to an HFHSD (at least 12 weeks) can increase blood glucose, insulin resistance, leptin levels, and potentially induce gestational diabetes, which could affect/bias measurements and interpretations. Do you have blood assays and potential glucose tolerance tests for your mice after this prolonged exposure to HFHSD?

For instance, leptin resistance in vagal afferent neurons inhibits cholecystokinin signaling and satiety in diet-induced obese animals (de Lartigue et al., 2012). Numerous studies showed that vagus nerve works differently in lean and obese states, with profound consequences on food intake.

It is known that the vagus nerve functions differently following prolonged exposure to an HFHSD (or HFD). The real original result of this work is that within HFHSD-mice, tension-sensitive GVA responses did not differ between non-pregnant and pregnant mice. This suggests that GVA desensitization to stretch due to gestation does not add to that obtained with HFHSD treatment. The real question is why? Through what mechanisms? How does nutrient sensing intervene?

To better understand the mechanisms involved in this "dual model" (gestation associated with stretch modulation and HFHSD associated with nutrient sensing modulation or adaptation), recording studies on the vagus nerve's response to different nutrients (sugar and fat) would have been insightful.

Minor Point:

Why did the authors use Glu-Venus mice, a strain marking L cells with fluorescence (GLP1-producing cells)? The rationale for using these mice is not clear. Have you worked with L cells? Please clarify in the text.

The role of GH receptor during gestation is intriguing and discussed in the paper. Have qPCR or RNAseq studies of the Nodose ganglia been conducted to support this hypothesis?

Have fluorescence recordings been conducted? One interesting experiment would be to record calcium activity in the Nodose Ganglia following stretch or nutrient exposure in pregnant mice (HFHSD or not).

While this study is intriguing, it feels somewhat frustrating as it lacks nutrient recordings and more insight into central-level information integration. Could you add some sentences about central signal integration in these 2 physiological states (obese and pregnant) and the combination of those?

Conclusion:

This work is interesting and complements the team's previous work; however, it is too observational (lacking mechanisms explanation) and solely under the prism of mechanosensitivity (major actor of satiety) which is quite limiting for the vagus nerve and control of food intake. Notwithstanding its focus solely on "stretch" and the absence of mechanistic approaches, the study is well-executed and provides a meaningful contribution to the scientific community.

END OF COMMENTS

Prof. Amanda Page
School of Biomedicine
University of Adelaide
Head: Vagal Afferent Research Group
Level 7, SAHMRI
North Terrace
Adelaide
Tel: (08) 8128 4840
E-mail: Amanda.page @adelaide.edu.au

Dear Editors,

Thank you for the opportunity to revise our paper “Pregnancy and a high-fat high-sugar diet each attenuate mechanosensitivity of murine gastric vagal afferents, with no additive effects”. We have edited the manuscript in response to feedback from the editor and reviewers. In particular, we have added new information about changes in hormone receptors in the nodose ganglia that are induced by pregnancy and high-fat high-sugar (HFHS) feeding. This provides information about a likely mechanism contributing to down-regulation of stretch responses in the stomach during pregnancy. Each specific comment is addressed below; line numbers refer to the tracked changes version of the manuscript.

We confirm that neither the manuscript nor parts of the content are under consideration or published in another journal.

We hope that you agree this manuscript is suitable for publication in Journal of Physiology.

Yours faithfully,

Prof Amanda Page on behalf of all authors

Responses to editor and reviewer

Reviewing Editor:

This is an interesting study by a group leader in the field. The study is well designed and well written. It demonstrates that GVA mechanosensitivity are altered by obesogenic diet during pregnancy does not alter changes induced by pregnancy observed in control mice. The effects of interactions between diet and obesity are novel. However, in its current form the study lacks mechanistic insights and remains largely descriptive.

Thank you for the feedback on our manuscript. We have performed additional measures on tissue collected from these mice (nodose ganglia) that we feel adds substantial mechanistic insight. Unfortunately, some additional measures suggested by reviewers are not possible as the tissue has already been used to complete the reported measures.

Major concerns:

1) It is not clear for the reviewer why was GVA sensitivity studied solely on one time point ie, almost at the end of the dynamic process of pregnancy. Additional time points, for studying GVA sensitivity should be provided. In particular based on the metabolic study, the period d 6,5 to 8,5 appears also to be of interest. Indeed, at this time point, both meal size as well as food intake are different between groups of interests. IN addition, at this time point body weight also appears different between P SLD and P HFHSD suggesting that changes in GVA sensitivity at this time could contribute to the increased weight gain of PSLD mice.

We thank the reviewing editor for this suggestion. Indeed, we have previously conducted a study similar to that proposed, where we measured responses of GVA tension receptors in non-pregnant, early pregnant, mid pregnant and late pregnant mice fed a standard laboratory diet (Li et al. 2012 Am J Physiol Gastro Liver Physiol 320:G183-192). In that study, changes in GVA responses to stretch became attenuated in mid-pregnancy, with much more substantial suppression at late-pregnancy. We therefore focussed the present study on GVA responses in late pregnancy. We have expanded the sentence at the end of the introduction that describes the study approach to include the rationale for the timing:

"We investigated this question using a mouse model of HFHSD feeding, focussing on late pregnancy when suppression of GVA responses to stretch is greatest (Li et al., 2021)." {lines 123 to 124 of marked copy}

2) From a mechanistical point of view, more data should be provided describing the interactions between diet and pregnancy upon the expression of known or candidate receptors or factors involved GVA mechanosensing during pregnancy and/or obesity. For instance, the expression of TRPV1, GLP1R or pregnancy associated hormones should be described to support the absence of difference.

Thank you for this suggestion. We have now measured and added information on expression of hormone receptors in the nodose ganglia, the location of the cell bodies of the vagal afferent nerve. We have now added data on expression of TRPV1, a known modulator of GVA responses to stretch,

as well as receptors for CCK, leptin, ghrelin and growth hormone (GH). We were excited to see up-regulation of the receptor for GH in the nodose during pregnancy, regardless of diet. Together with our previous report of GH-induced down-regulation of GVA responses to stretch (Li et al. 2012 Am J Physiol Gastro Liver Physiol 320:G183-192), this implicates increasing circulating GH concentrations during murine pregnancy (Gatford et al. 2017 Endocr Connections 6:260-266) as likely mechanisms underlying down-regulated GVA responses to stretch during pregnancy in the mouse. We have added information of RT-PCR of nodose ganglia to the methods section {lines 201 to 220} and results {lines 506 to 511} for receptor expression to text and Figure 5. The discussion has been expanded to include:

“Dampened tension-sensitive GVA mechanosensitivity during pregnancy is likely driven by changes in hormone levels (Clarke et al., 2021) or their signalling pathways. In the whole nodose ganglia, mRNA expression of receptors that regulate vagal afferent sensitivity, including TRPV1 (Kentish et al., 2015a), CCKA (Daly et al., 2011), GSHR (Kentish et al., 2012) and LepR (Kentish et al., 2013b), were unaffected by pregnancy suggesting they are not involved in the observed changes in GVA mechanosensitivity during pregnancy in SLD-mice. In contrast, expression of GHR in the nodose ganglia was increased in pregnancy regardless of diet. We have previously reported that ex vivo administration of growth hormone (GH) decreases the response of murine tension-sensitive GVAs to stretch (Li et al., 2021), and that circulating GH increases from early- to mid-pregnancy and then remains elevated during late-pregnancy in mice (Gatford et al., 2017). Neuron-specific GHR ablation reduced food intake in pregnant mice suggesting that at least part of pregnancy-induced hyperphagia is driven by central GHR signalling (Teixeira et al., 2019). Together, this suggests GH down-regulates peripheral as well as central satiety pathways to permit increased food intake during pregnancy.” {lines 548 to 566}

“Although nodose ganglia GHR expression was upregulated during pregnancy regardless of diet, GHR-mediated down-regulation of GVA responses to stretch may be impaired in obesity due to lower circulating GH abundance. In women, circulating concentrations of the placental GH variant are negatively correlated with maternal body mass index during pregnancy (Coutant et al., 2001; Verhaeghe et al., 2002). Whether obesity similarly down-regulates circulating GH abundance in species where GH remains pulsatile during pregnancy and is not secreted by the placenta, like mice, have not been evaluated. Nevertheless, obesity down-regulates circulating GH in non-pregnant mice, as in non-pregnant humans (Steyn et al., 2013).” {lines 641 to 649}

3) In fig 1, it is not clear why there is no difference in body weight gain is observed between P SLD and P HFHSD as at the beginning of mating HFHSD mice have a greater body weight than SLD mice?

Variation in initial weight as well as weight at study day 17 within each of these groups was such that the total weight gain during the study did not differ (Fig. 1B).

Senior Editor

The editor apologizes sincerely for the significant delays that occurred during the review of this manuscript. This resulted from challenges in identifying willing reviewers. The Reviewing Editor conducted an independent review and indicates that the manuscript, while interesting, lacks mechanistic information. The external reviewer actually suggests some additional mechanistic studies that could be performed. In light of the delays that have occurred, I would like to offer you the opportunity to revise your manuscript if additional data can be generated. Alternatively, we

can offer to refer your manuscript to Experimental Physiology. No matter what, thank you for submitting your work to JP.

Thank you for the opportunity to revise the manuscript and resubmit to Journal of Physiology.

Precise p values must be supplied for all statistical comparisons.

We have revised the results text, Table 1 and figures or figure legends to provide exact p values for all comparisons, including p values for post-hoc analyses on Figures 1B, and 3 (all panels). Because Figures 1A and 2 show time-courses, we feel that adding exact p-values for each comparison would make these figures very difficult to read. We have therefore submitted a link to tables summarising the p values for each outcome and post-hoc comparison and request this link be provided to readers as supplementary material. This link also provides exact p values for all contrasts assessed in Figure 4B. The link is provided at the end of figure legend 1, 2 and 4.

Referee #1

In the study titled "Pregnancy and a high-fat high-sugar diet each attenuate mechanosensitivity of murine gastric vagal afferents, with no additive effects," Georgia Clarke and her collaborators explored the capacity of gastric vagal afferents (GVA) to modulate their mechanosensitive response under two physiological conditions: gestation and exposure to a high-fat high-sugar diet (HFHSD), both individually and in combination.

The observational results are well-executed in metabolic cage and vagal nerve electrophysiology during stretching. While the results are interesting and support previous findings, they seem overly observational and focus too much on stretching, a crucial aspect of satiety, but neglect nutrient sensing especially when dealing with HFHSD, and potential obese state.

Indeed, vagal afferent response to stretch correlates with meal size (Li et al., 2021), as validated in this manuscript. However, as suggested by the authors, vagal response is not solely dependent on stretching; there is a nutrient-dependent response, especially to fats and sugars activating gastric vagal afferents.

Thank you for your feedback. Indeed, we agree that upregulation of daily food intake during pregnancy in mice fed a standard laboratory diet, but not in those fed a HFHS diet, likely reflects nutrient signalling, since changes in gastric vagal afferent responses do not explain differential feed intake between diet groups in pregnancy. Indeed, this most likely reflects impacts of high dietary fat, which induces strong satiety responses (Maher & Clegg 2019 Critical Reviews in Food Science and Nutrition **59**, 1619-1644). This could be at the level of vagal afferents from the intestine as well as impacts on central satiety. We have discussed the potential mechanisms for effects of nutrients on satiety in the manuscript {lines 602 to 612}. We agree that our results suggest additional studies of satiating effects of nutrients in pregnancy and in response to HFHS-feeding would be merited but beyond the scope of the current study.

Major points:

Prolonged exposure to an HFHSD (at least 12 weeks) can increase blood glucose, insulin resistance, leptin levels, and potentially induce gestational diabetes, which could affect/bias measurements and interpretations. Do you have blood assays and potential glucose tolerance tests for your mice after this prolonged exposure to HFHSD?

Thank you for this suggestion. The metabolic phenotype of these mice has indeed been characterised in detail before, during and after pregnancy, and these outcomes are the subject of a separate paper.

For instance, leptin resistance in vagal afferent neurons inhibits cholecystokinin signaling and satiety in diet-induced obese animals (de Lartigue et al., 2012). Numerous studies showed that vagus nerve works differently in lean and obese states, with profound consequences on food intake.

It is known that the vagus nerve functions differently following prolonged exposure to an HFHSD (or HFD). The real original result of this work is that within HFHSD-mice, tension-sensitive GVA responses did not differ between non-pregnant and pregnant mice. This suggests that GVA desensitization to stretch due to gestation does not add to that obtained with HFHSD treatment. The real question is why? Through what mechanisms? How does nutrient sensing intervene?

We agree that impacts of obesity on vagal afferents have been reported previously. However, this is the first study investigating the impact of a high fat high sugar diet on vagal afferent sensitivity and is the first study to investigate the responses in the context of pregnancy, important since half the women in developed countries enter pregnancy obese.

Our new data showing upregulation of the GHR in the nodose ganglia during pregnancy, together with evidence that GH increases during pregnancy and that GH down-regulates gastric vagal stretch responses in non-pregnant female mice, suggests that increased GH signalling may underlie down-regulation of gastric stretch responses during pregnancy. The available data, however, suggests that responses to GH do not underlie impacts of HFHSD-feeding. This has been discussed in the revised manuscript {lines 548 to 566 and lines 641 to 649}

To better understand the mechanisms involved in this "dual model" (gestation associated with stretch modulation and HFHSD associated with nutrient sensing modulation or adaptation), recording studies on the vagus nerve's response to different nutrients (sugar and fat) would have been insightful.

We agree that studies of pregnancy and HFHSD on vagal nerve responses to nutrients would be interesting. Testing these *ex vivo*, in gastric preparations, is unlikely to be informative, however, as nutrient-driven satiety responses are primarily evoked by receptors in the small intestine. We plan to study intestinal vagal responses in future.

Minor Point:

Why did the authors use Glu-Venus mice, a strain marking L cells with fluorescence (GLP1-producing cells)? The rationale for using these mice is not clear. Have you worked with L cells? Please clarify in the text.

The mice in the present study were part of a larger study, and use of this strain allows separation of alpha- and beta-cells within pancreatic cell populations. We have added an explanation of the choice of strain to the methods section as requested:

"The mice in this study were part of a larger study where use of this strain allowed separation of alpha- and beta-cells within pancreatic cell populations." {lines 134 to 136}

The role of GH receptor during gestation is intriguing and discussed in the paper. Have qPCR or RNAseq studies of the Nodose ganglia been conducted to support this hypothesis?

As suggested, we have performed qPCR studies in nodose ganglia collected from the present cohort of mice. Providing further evidence to support the role of growth hormone in vagal afferent-driven gastric responses, nodose ganglion expression of the growth hormone receptor was upregulated during pregnancy, regardless of diet. We have added information of RT-PCR of nodose ganglia to the methods section {lines 201 to 220} and results {lines 506 to 511} for receptor expression to text and Figure 5. Further, information has been added to the discussion {lines 548 to 566 and lines 641 to 649}.

Have fluorescence recordings been conducted? One interesting experiment would be to record calcium activity in the Nodose Ganglia following stretch or nutrient exposure in pregnant mice (HFHSD or not).

Thank you for this suggestion. The approach used in the present study allows us to record vagal afferent responses to stretch *ex vivo*, but we agree that an *in vivo* approach would be useful in future studies particularly of nutrient responses.

While this study is intriguing, it feels somewhat frustrating as it lacks nutrient recordings and more insight into central-level information integration. Could you add some sentences about central signal integration in these 2 physiological states (obese and pregnant) and the combination of those?

This article focuses on peripheral satiety signalling and, as such, the discussion has been limited to this. However, we have added information on GH central satiety signalling given that GH appears to down-regulate peripheral as well as central satiety pathways during pregnancy:

“Neuron-specific GHR ablation reduced food intake in pregnant mice suggesting that at least part of pregnancy-induced hyperphagia is driven by central GHR signalling (Teixeira et al., 2019). Together, this suggests GH down-regulates peripheral as well as central satiety pathways to permit increased food intake during pregnancy.” {lines 559 to 566}

Conclusion:

This work is interesting and complements the team's previous work; however, it is too observational (lacking mechanisms explanation) and solely under the prism of mechanosensitivity (major actor of satiety) which is quite limiting for the vagus nerve and control of food intake. Notwithstanding its focus solely on "stretch" and the absence of mechanistic approaches, the study is well-executed and provides a meaningful contribution to the scientific community.

Thank you for your comment. We agree that gastric vagal afferents play an important role in a cascade of different satiety signals that originate from the gastrointestinal tract. Small intestinal afferents play another important role which will form part of future studies. As stated above, we have now measured and added information on expression of hormone receptors in the nodose ganglia. We have shown up-regulation of the receptor for GH in the nodose during pregnancy,

regardless of diet. Together with our previous report of GH-induced down-regulation of GVA responses to stretch (Li et al. 2012 Am J Physiol Gastro Liver Physiol 320:G183-192), this implicates increasing circulating GH concentrations during murine pregnancy (Gatford et al. 2017 Endocr Connections 6:260-266) as likely mechanisms underlying down-regulated GVA responses to stretch during pregnancy in the mouse. We have added information of RT-PCR of nodose ganglia to the methods section {lines 201 to 220}, results {lines 506 to 511 and Figure 5} and the discussion.

Dear Dr Page,

Re: JP-RP-2024-286115R1 "Pregnancy and a high-fat high-sugar diet each attenuate mechanosensitivity of murine gastric vagal afferents, with no additive effects." by Georgia S Clarke, Hui Li, Elaheh Heshmati, Lisa M Nicholas, Kathryn L Gatford, and Amanda J Page

Thank you for submitting your manuscript to The Journal of Physiology. It has been assessed by a Reviewing Editor and by 1 expert referee and we are pleased to tell you that it is acceptable for publication following satisfactory revision.

REVISION CHECKLIST:

We look forward to receiving your revised submission.

Yours sincerely,

Kim Barrett
Senior Editor
The Journal of Physiology

EDITOR COMMENTS

Thank you for your significant efforts in revising the manuscript. The reviewer is only suggesting one substantive additional revision - to acknowledge any possible relationship between metabolic status and responses to stress. In the view of the senior editor, this would not necessarily require the provision of additional data unless these are already available to the authors, but could rather be handled by additional text to acknowledge the points.

REFEREE COMMENTS

Referee #1:

We thank the authors for addressing the majority of the reviewers' comments and for their thoughtful responses to the raised questions. The manuscript, in its revised form, presents novel findings and contributes valuable insights to the field. Notably, this study is the first to investigate the impact of a high-fat, high-sugar diet on vagal afferent sensitivity and to examine these responses in the context of pregnancy.

However, we encountered significant challenges in following the authors' corrections and additions due to discrepancies between the citations, added text, and the line numbers provided in the revised document. It would be greatly appreciated if these could be aligned to ensure clarity and facilitate the review process.

Regarding the responses and data provided:

- The inclusion of qPCR results adds valuable confirmation of the potential mechanisms involved. The authors effectively suggest that GH downregulates both peripheral and central satiety pathways to allow increased food intake during pregnancy. The findings that mRNA expression of receptors regulating vagal afferent sensitivity (e.g., TRPV1, CCKA, GSHR, and LepR) remained unchanged in pregnancy, while GHR expression increased, offer compelling evidence supporting their hypothesis. This is further supported by prior studies implicating GH in modulating GVA responses during murine pregnancy.
- Nevertheless, to better understand the mechanisms at play, it would be crucial to include the metabolic status of the mice, particularly its role in GVA mechanosensitivity during pregnancy, both with and without the high-fat, high-sugar diet. This would provide a more comprehensive understanding of the observed effects.

Given the constraints of publication timelines, we understand that additional data may be challenging to provide at this stage. However, we strongly encourage the authors to include a well-referenced and detailed discussion of the potential correlation between the metabolic status of the animals and the observed GVA responses to stretch. Additionally, if metabolic data are already available, we strongly suggest their inclusion along with an appropriate discussion of their implications.

In summary, this manuscript, incorporating the above suggestions, represents an important and novel contribution to the scientific community. We believe it is suitable for publication in The Journal of Physiology, pending the alignment of references, line numbers, and a more detailed exploration of the metabolic aspects as discussed.

END OF COMMENTS

Prof. Amanda Page
School of Biomedicine
University of Adelaide
Head: Vagal Afferent Research Group
Level 7, SAHMRI
North Terrace
Adelaide
Tel: (08) 8128 4840
E-mail: Amanda.page @adelaide.edu.au

Dear Editors,

Thank you for the opportunity to revise our paper “Pregnancy and a high-fat high-sugar diet each attenuate mechanosensitivity of murine gastric vagal afferents, with no additive effects”. We have edited the manuscript in response to feedback from the editor and reviewers. Each specific comment is addressed below; line numbers refer to the tracked changes version of the manuscript.

We confirm that neither the manuscript nor parts of the content are under consideration or published in another journal.

We hope that you agree this manuscript is suitable for publication in Journal of Physiology.

Yours faithfully,

Prof Amanda Page on behalf of all authors

Response to editor comments

Thank you for your significant efforts in revising the manuscript. The reviewer is only suggesting one substantive additional revision - to acknowledge any possible relationship between metabolic status and responses to stress. In the view of the senior editor, this would not necessarily require the provision of additional data unless these are already available to the authors, but could rather be handled by additional text to acknowledge the points.

We have now added additional text to the discussion as suggested:

“It is therefore possible the obese state rather than the diet induced the previously reported changes in diurnal food intake patterns. However, this study formed part of a larger cohort study where HFHSD feeding resulted in reduced insulin sensitivity, hyperinsulinemia and impaired glucose tolerance (O'Hara et al., 2023) and, therefore, there are still diet-induced metabolic changes that may impact on circadian patterns of food intake patterns.” (Marked copy, page 25, lines 577-581)

“Alternatively, given the reduced insulin sensitivity, hyperinsulinemia and impaired glucose tolerance of pregnant HFHSD mice (O'Hara et al., 2023) it is possible that elevated post-prandial glucose might also contribute to reduced GVA mechanosensitivity. Similar patterns of reduced insulin sensitivity and impaired glucose tolerance are also seen during non-obese mouse and human pregnancy (Lain & Catalano, 2007; Musial et al., 2016; Powe et al., 2019). However, whether elevated glucose contributes in pregnancy and HFHSD-fed animals to suppress GVA mechanosensitivity requires further investigation since the impact of elevated glucose on GVA mechanosensitivity has not been reported.” (Marked copy, page 26, lines 598-606)

Response to referee 1

We thank the authors for addressing the majority of the reviewers' comments and for their thoughtful responses to the raised questions. The manuscript, in its revised form, presents novel findings and contributes valuable insights to the field. Notably, this study is the first to investigate the impact of a high-fat, high-sugar diet on vagal afferent sensitivity and to examine these responses in the context of pregnancy.

We thank the reviewer for the important comments raised.

However, we encountered significant challenges in following the authors' corrections and additions due to discrepancies between the citations, added text,

and the line numbers provided in the revised document. It would be greatly appreciated if these could be aligned to ensure clarity and facilitate the review process.

We apologise for any confusion. The page and line numbers provided were from the marked copy of the manuscript.

The inclusion of qPCR results adds valuable confirmation of the potential mechanisms involved. The authors effectively suggest that GH downregulates both peripheral and central satiety pathways to allow increased food intake during pregnancy. The findings that mRNA expression of receptors regulating vagal afferent sensitivity (e.g., TRPV1, CCKA, GSHR, and LepR) remained unchanged in pregnancy, while GHR expression increased, offer compelling evidence supporting their hypothesis. This is further supported by prior studies implicating GH in modulating GVA responses during murine pregnancy.

We agree with this comment.

Nevertheless, to better understand the mechanisms at play, it would be crucial to include the metabolic status of the mice, particularly its role in GVA mechanosensitivity during pregnancy, both with and without the high-fat, high-sugar diet. This would provide a more comprehensive understanding of the observed effects.

Given the constraints of publication timelines, we understand that additional data may be challenging to provide at this stage. However, we strongly encourage the authors to include a well-referenced and detailed discussion of the potential correlation between the metabolic status of the animals and the observed GVA responses to stretch. Additionally, if metabolic data are already available, we strongly suggest their inclusion along with an appropriate discussion of their implications.

As stated previously, this study forms part of a larger study that examined the metabolic status of these mice. This manuscript is currently being prepared for publication. Nonetheless, as suggested we have expanded the discussion, including reference to a published abstract that provides detail on the metabolic status of these mice.

“It is therefore possible the obese state rather than the diet induced the previously reported changes in diurnal food intake patterns. However, this study formed part of a larger cohort study where HFHSD feeding resulted in reduced insulin sensitivity, hyperinsulinemia and impaired glucose tolerance (O'Hara et al., 2023) and, therefore,

there are still diet-induced metabolic changes that may impact on circadian patterns of food intake patterns.” (Marked copy, page 25, lines 577-581)

“Alternatively, given the reduced insulin sensitivity, hyperinsulinemia and impaired glucose tolerance of pregnant HFHSD mice (O'Hara et al., 2023) it is possible that elevated post-prandial glucose might also contribute to reduced GVA mechanosensitivity. Similar patterns of reduced insulin sensitivity and impaired glucose tolerance are also seen during non-obese mouse and human pregnancy (Lain & Catalano, 2007; Musial et al., 2016; Powe et al., 2019). However, whether elevated glucose contributes in pregnancy and HFHSD-fed animals to suppress GVA mechanosensitivity requires further investigation since the impact of elevated glucose on GVA mechanosensitivity has not been reported.” (Marked copy, page 26, lines 598-606)

Dear Professor Page,

Re: JP-RP-2025-286115R2 "Pregnancy and a high-fat high-sugar diet each attenuate mechanosensitivity of murine gastric vagal afferents, with no additive effects." by Georgia S Clarke, Hui Li, Elaheh Heshmati, Lisa Marie Nicholas, Kathryn L Gatford, and Amanda J Page

We are pleased to tell you that your paper has been accepted for publication in The Journal of Physiology.

Yours sincerely,

Kim Barrett
Senior Editor
The Journal of Physiology

If you would like to receive our 'Research Roundup', a monthly newsletter highlighting the cutting-edge research published in The Physiological Society's family of journals (The Journal of Physiology, Experimental Physiology, Physiological Reports, The Journal of Nutritional Physiology and The Journal of Precision Medicine: Health and Disease), please click this link, fill in your name and email address and select 'Research Roundup':
<https://www.physoc.org/journals-and-media/membernews>

- **TRANSPARENT PEER REVIEW POLICY:** To improve the transparency of its peer review process, The Journal of Physiology publishes online as supporting information the peer review history of all articles accepted for publication. Readers will have access to decision letters, including Editors' comments and referee reports, for each version of the manuscript as well as any author responses to peer review comments. Referees can decide whether or not they wish to be named on the peer review history document.
- You can help your research get the attention it deserves! Check out Wiley's free Promotion Guide for best-practice recommendations for promoting your work at: www.wileyauthors.com/eo/guide. You can learn more about Wiley Editing Services which offers professional video, design, and writing services to create shareable video abstracts, infographics, conference posters, lay summaries, and research news stories for your research at: www.wileyauthors.com/eo/promotion.
- **IMPORTANT NOTICE ABOUT OPEN ACCESS:** To assist authors whose funding agencies mandate public access to published research findings sooner than 12 months after publication, The Journal of Physiology allows authors to pay an Open Access (OA) fee to have their papers made freely available immediately on publication.

REFeree COMMENTS

Referee #1:

We appreciate the thorough review process and the valuable feedback provided. The modifications made to the manuscript throughout this process have been substantial and have significantly strengthened the study, making it even more relevant and insightful for the scientific community.

The findings presented in this study contribute to our understanding of how gastric vagal afferents regulate food intake, particularly in the context of pregnancy and HFdiet. While the absence of metabolic correlation is indeed a limitation, we acknowledge that this aspect could be addressed in a broader study specifically designed to explore these metabolic dimensions.

Overall, we find that the revised manuscript is now robust and satisfactory, adequately addressing the key scientific questions raised during the review process.